# *Drosophila* as a Model for Infectious Diseases

**DOI:** 10.3390/ijms22052724

**Published:** 2021-03-08

**Authors:** J. Michael Harnish, Nichole Link, Shinya Yamamoto

**Affiliations:** 1Department of Molecular and Human Genetics, Baylor College of Medicine (BCM), Houston, TX 77030, USA; Jacob.Harnish@bcm.edu (J.M.H.); nichole.link@neuro.utah.edu (N.L.); 2Jan and Dan Duncan Neurological Research Institute, Texas Children’s Hospital, Houston, TX 77030, USA; 3Howard Hughes Medical Institute, Houston, TX 77030, USA; 4Department of Neuroscience, BCM, Houston, TX 77030, USA; 5Development, Disease Models and Therapeutics Graduate Program, BCM, Houston, TX 77030, USA

**Keywords:** *Drosophila melanogaster*, immunity, infection, pathogens and virulence factors, disease models

## Abstract

The fruit fly, *Drosophila melanogaster*, has been used to understand fundamental principles of genetics and biology for over a century. *Drosophila* is now also considered an essential tool to study mechanisms underlying numerous human genetic diseases. In this review, we will discuss how flies can be used to deepen our knowledge of infectious disease mechanisms in vivo. Flies make effective and applicable models for studying host-pathogen interactions thanks to their highly conserved innate immune systems and cellular processes commonly hijacked by pathogens. *Drosophila* researchers also possess the most powerful, rapid, and versatile tools for genetic manipulation in multicellular organisms. This allows for robust experiments in which specific pathogenic proteins can be expressed either one at a time or in conjunction with each other to dissect the molecular functions of each virulent factor in a cell-type-specific manner. Well documented phenotypes allow large genetic and pharmacological screens to be performed with relative ease using huge collections of mutant and transgenic strains that are publicly available. These factors combine to make *Drosophila* a powerful tool for dissecting out host-pathogen interactions as well as a tool to better understand how we can treat infectious diseases that pose risks to public health, including COVID-19, caused by SARS-CoV-2.

## 1. Introduction

Widespread vaccinations, the development of antibiotics, and increased hygiene standards have all reduced the burden of infectious diseases. Despite this, infectious diseases caused by viruses, bacteria, fungi, and parasites still remain leading causes of death worldwide [1]. Additionally, new pathogens constantly emerge or evolve to avoid our current treatments, as evident from the ongoing COVID-19 pandemic caused by SARS-CoV-2 [2]. There is continuous work that must be done by the medical and scientific community to ensure treatments keep up with evolving diseases, work that relies on our ability to understand the precise molecular mechanisms utilized by pathogens to take advantage of the host. Understanding how a pathogen spreads, evades host immunity, reprograms cellular machinery, and ultimately causes damage to the host allows us to design strategies to more effectively prevent or treat these diseases. Such knowledge is crucial for the development of new antibiotics and antiviral drugs to combat resistant infections as well as treating novel pathogens as they appear.

Most studies involving host-pathogen interactions have been performed in cultured mammalian cells and using in vivo murine models. This is understandable as these systems effectively model certain aspects of the host environment in humans. Pathogens often affect biological pathways that are highly conserved throughout evolution, including components that are involved in innate immunity such as the NF-κB (Nuclear Factor kappa B) and JNK (c-Jun N-terminal Kinase) signaling pathways, as well as fundamental cellular processes such as phagocytosis and apoptosis. Due to this, non-mammalian multicellular organisms such as nematode worms, fruit flies, and zebrafish can be used alongside mammalian model systems to study host-pathogen interactions in a complimentary fashion. This is beneficial as these model systems are amenable to powerful and sophisticated genetic manipulations [3].

The fruit fly, *Drosophila melanogaster,* has been used as a model organism to understand fundamental mechanisms of genetics and development due to its rapid lifecycle, cost-effectiveness, and available advanced technologies [4]. More recently, flies have emerged as useful tools to study human diseases such as rare Mendelian diseases [5], neurodegenerative disorders [6], and cancer [7]. *Drosophila* is also an attractive model system to assess molecular mechanisms of pathogenic proteins encoded in the genomes of viral and bacterial genomes for several reasons. First, flies have an innate immune system that responds to foreign pathogens by activating cellular pathways to produce antimicrobial peptides, promote inflammation and recruit further immune system players, including hemocytes that have phagocytic capacity. The innate immune system is highly conserved, including the ways in which it detects and responds to pathogens and the genes involved in these processes [8]. This is most evident through the fact that studies on the fly *toll* mutant led to the identification of the Toll-like receptor signaling pathway in mammals [9], a discovery that was recognized through the 2011 Nobel Prize in Physiology or Medicine to Dr. Jules Hoffman. Second, fruit flies are, genetically speaking, highly tractable. This allows researchers to conduct complicated experiments in vivo [10,11]. For example, tissue and time-specific expression of a specific gene, or groups of genes, can easily be performed using the UAS/GAL4 system, described in detail by Brand and Perrimon [12]. The UAS/GAL4 system allows controlled spatiotemporal expression of genes engineered with an upstream UAS (Upstream Activation Sequence) sequence. Flies harboring the UAS transgene are then crossed to GAL4 lines in which the GAL4 transcriptional activator is expressed under the control of a specific gene promoter (e.g., GMR to drive expression in the developing eye). Since GAL4 activates the transcription of the gene downstream of UAS, any cell type in which the GAL4 is expressed will also express a transgene under UAS control [12]. More recently, an expanding array of numerous techniques have provided fly geneticists with additional tools to activate or inactivate any gene in any cell type at any time point [13,14,15]. Genetic interaction and epistasis analyses that require simultaneous manipulation of multiple chromosomes can also be achieved rapidly and easily compared to vertebrate model organisms. Third, *Drosophila* also have well documented and profiled morphological phenotypes that can act as rapid readouts of certain cellular processes. For example, morphological defects in the adult wing often indicates defects in evolutionarily conserved developmental signaling pathways, including the Notch, Wnt, Hedgehog, BMP/TGF-β (Bone Morphogenic Protein/Transforming Growth Factor beta), and RTK (Receptor Tyrosine Kinase) signaling pathways [16]. Hence, in addition to assessing the function of pathogenic proteins using the fly immune system, one can explore the impact they may have on other tissue types to extract fundamental knowledge about their molecular functions. Fourth, public stock centers, including the Bloomington *Drosophila* Stock Center [17] and Kyoto Stock Center [18], collect and distribute many different types of genetic reagents from and to the community, providing easy access to various useful tools that can be quickly utilized. In summary, readily available reagents, sophisticated genetic manipulations, rapid experimentation, and a well-conserved innate immune system all converge to make *Drosophila* a powerful tool for studying infectious diseases [19].

There are currently two prevalent methods to study infection using *Drosophila*; direct infection and ectopic expression of pathogenic proteins [20,21,22]. The first method actively infects the fly with the pathogen of interest via feeding or microinjections [23]. The benefit of this method is that it allows researchers to study host-pathogen interactions on a whole animal scale, and one can also study the effect of pathogens that do not infect *Drosophila melanogaster* in the wild. The second more widespread approach investigates host-pathogen interactions by ectopically over-expressing a pathogenic protein of interest with the UAS/GAL4 system. In this method, one can control the timing and tissue-specific expression of a protein of interest by cloning the pathogenic protein of interest under the UAS and a minimal promotor and driving its expression using a GAL4 transcriptional activator driven by a specific enhancer. This allows one to study the effects of a single virulence factor at a time in a tissue of interest. It is also possible to co-express several proteins at a time to study proteins that may function together. Virulence factors can also be driven in different genetic backgrounds (e.g., knockouts) to help determine which host proteins the pathogenic proteins are physically and/or genetically interacting with. Finally, transgenic flies can be made which express single domains of pathogenic proteins or express pathogenic proteins lacking possible regulatory domains to perform structure-function analysis in vivo. In addition to studying viral, bacterial, and fungal proteins, several studies have investigated the effects of prion proteins using *Drosophila* [24]. It is important to note that these two methods, direct infection and exogenous expression of virulence factors, are not mutually exclusive and are often used in combination. Finally, both models can be used to screen for drugs that help combat infectious diseases [25], making *Drosophila* a model organism for translational biomedical sciences.

In this review, we will primarily focus on studies that have utilized the ectopic over-expression method to study host-pathogen interactions in vivo using fruit flies (Table 1). We will first introduce several biological pathways in which virulence factors have been shown to have an impact. Next, we will discuss different virulence factors that have been explored in *Drosophila* and summarize the discoveries that have been made using fruit flies. Finally, we will discuss several studies that have utilized *Drosophila* to identify potential drugs that can be translated to clinics.

## 2. Innate Immune Signaling Pathways and Pathogenic Proteins that Affect Them

Originally discovered in mammalian B cells [26], NF-κB is a protein complex that plays key roles in innate immunity that are highly conserved in evolution [27]. In *Drosophila*, the NF-κB pathway primarily responds to infection by stimulating the downstream production of antimicrobial peptides (AMPs) in the fat body, an organ that carries out functions mediated by the liver and adipose tissue in mammals [28]. While there are no direct orthologs of *Drosophila* AMP genes in humans, a number of human peptides that have antimicrobial activity have been identified [29]. There are three genes in the fly genome that encode NF-κB proteins; *dorsal*, *DIF* (*Dorsal-related immunity factor*), and *relish* [30]. NF-κB proteins are found at rest within the cytoplasm bound by inhibitory IκB (Inhibitor of NF-κB) proteins. For Dorsal and DIF, a protein encoded by the *cactus* gene functions as their IκB. For Relish, the C-terminal portion of this protein (Rel-49) functions as an IκB that binds to the N-terminal portion of this protein (Rel-68) to keep it inactive. Upon stimulation by a pathogen, the IκB proteins are phosphorylated by the IκB kinase (IKK, encoded by IKKβ, IKKε and IKKγ in flies) complex, ubiquitinated by β-TRcP (beta-transducin repeat containing E3 ubiquitin protein ligase, encoded by the *supernumerary limbs* (*slmb*) gene in flies), and ultimately degraded [31]. For Relish, a proteolytic cleavage mediated by the caspase Dredd is also necessary to release Rel-49 from Rel-68 [32]. After being released from IκB, Dorsal, DIF, and Rel-68 translocate to the nucleus to activate transcription of key immune genes, including AMP encoding genes. Activation of the NF-κB pathway can occur through one of the two molecularly distinct signaling pathways: Toll and IMD (Immune deficiency) (Figure 1). The Toll pathway is activated in response to both gram-positive bacterial or fungal pathogens, while the IMD pathway is activated in response to gram-negative bacteria. The end result of both pathways is the production of a unique set of AMPs to combat the pathogen. The Toll pathway produces Drosomycin while the IMD pathway induces the production of Diptericins [33]. In fact, differential expression of these genes can serve as convenient readouts to determine which immune signaling pathway is activated or affected by the virulence factor of interest. Certain pathogens have evolved several strategies to disrupt the NF-κB cascade to evade being attacked by the immune system. Some evasion methods specifically affect the Toll or IMD pathway, whereas others act on both pathways simultaneously. These disruptions not only help the pathogen survive but can also weaken the host to become susceptible to further pathogenic insults.

In addition to Toll and IMD pathways, the JNK pathway also plays an important role in the innate immunity of *Drosophila* [34] (Figure 2). JNK is a MAPK (Mitogen-Activated Protein Kinase) cascade that can become activated upon infection. The JNK cascade activates a set of genes to mediate inflammatory responses in both flies [35] and in mammals [36]. Numerous other stressors can activate this pathway, meaning it is often associated with cell survival and apoptosis in diverse contexts [37]. MAPK signaling pathways, which include JNK, ERK (extracellular signal-regulated kinase), and p38, involve activation of three core kinases, which are often generically referred to as the MAPK, MAPK kinase (MAPKK), and MAPKK kinase (MAPKKK) [38]. The JNK cascade in *Drosophila* starts from JNKKKs (JNK Kinase Kinases), which are encoded by multiple genes, including *slipper* (also known as *MLK*), *Tak1 (TGF-β activated kinase 1)*, and *Mekk1* (*Mitogen-activated protein kinase kinase kinase 1*) [39]. These kinases, in turn, act on downstream JNKKs (JNK Kinases) encoded by *MKK4* (*Mitogen-activated protein Kinase Kinase 4*) and *hemipterous (hep)* genes. JNK, the final kinase in the cascade, is encoded by the *basket* (*bsk*) gene. Bsk phosphorylates transcription factors such as Foxo (forkhead box subgroup O) and AP-1 (heterodimer of c-Jun and c-Fos), which, in turn, mediate transcriptional changes within the cell. Perturbations to this pathway often affect cell survival and can be either anti-apoptotic or pro-apoptotic, depending on the context. It is important to note that these key pathways involved in immunity are able to work in concert with other pathways, including JAK-STAT (Janus kinase-Signal Transducer and Activator of Transcription) signaling and Notch signaling (see Section 4 of this article) to make complex decisions about cell fate upon infection [40,41]. In this section, we will discuss the various ways in which specific pathogenic proteins disrupt host immunity and how flies have been used to study these effects.

### 2.1. Salmonellae Enterica: AvrA and NF-κB (IMD)/JNK Signaling

*Salmonellae enterica* is a gram-negative bacterium most commonly associated with salmonellosis; a condition contracted via consumption of infected food or water and characterized by fever, abdominal pain, and diarrhea [42]. This pathogen can enter into the bloodstream from the digestive tract and can be fatal. Previous work on a pathogenic *Salmonella* serotype (*S. typhimurium*) demonstrated that an effector protein, AvrA, was key in suppressing immune mechanisms in human cells [43]. *AvrA* encodes a secreted acetyltransferase that shares homology to *YopJ*, an immune effector from a different pathogenic bacteria family, *Yersinia*. *Drosophila* models were developed to study the molecular functions of AvrA in vivo [44]. The authors of this study created two transgenic strains of flies in which they expressed either wild-type AvrA or a catalytically dead form. Wild-type AvrA expression in the fat body prevented the IMD pathway from properly activating. However, the catalytically dead AvrA did not affect IMD activation, indicating that the acetyltransferase function of this enzyme is critical in regulating immunity. Interestingly, AvrA expression inhibited the translocation of Relish to the nucleus while DIF was unaffected (Figure 1). These results explain why the AvrA affects the IMD pathway but not the Toll pathway. While over-expression of AvrA alone did not affect lifespan, it did reduce the fly’s ability to respond to subsequent infections with gram-positive bacteria or fungi, suggesting these animals were immunocompromised. The authors also noted that AvrA expression led to a reduction in MKK4 (JNKK) activity, a key kinase in the JNK pathway (Figure 2). The involvement of AvrA in JNK signaling was further supported by genetic interaction experiments performed in the eye. Expression of constitutively active Eiger (a JNK pathway ligand) or Tak1 (JNKKK) causes rough eye phenotypes via excessive cell death due to hyperactivation of JNK signaling. Co-expression of AvrA was able to suppress this phenotype, indicating that this protein acts downstream of these factors. These data suggest AvrA can function as an anti-apoptotic virulence factor in addition to modulating the IMD branch of the innate immune pathway in vivo. Disruptions to these pathways likely prevent cytokine production and apoptosis of the host cell, aiding the survival of the bacteria during the initial stages of *S. typhimurium* infection. Interestingly, follow-up work in mouse models also demonstrated the importance of AvrA in *Salmonella* infection as AvrA-deficient strains of *S. typhimurium* caused increased cytokine production and increased apoptosis of macrophages [45]. This stronger immune response led to worsening of the course of the disease, indicating AvrA may have evolved to dampen immune responses to allow the pathogen to survive longer within its host. In another study using human intestinal epithelial cells, AvrA from *S. enterica* was also shown to stabilize tight junctions via suppressing JNK activity to reduce bacterial invasion and support host survival [46]. These findings provide an interesting view that although virulence factors can prevent proper immune functions in the host to increase their pathogenicity, some factors may actually benefit the host by smoothing the course of the infection to maintain a suitable environment for the pathogen to reproduce longer.

### 2.2. Aeromonas Salmonicida: AopP and NF-κB (Toll and IMD) Signaling

*Aeromonas salmonicida* is a gram-negative bacterium that is primary found in salmon and other fresh water fishes [47]. Infection with *A. salmonicida* is primarily known to cause furunculosis, hemorrhage, and sepsis in fish, but it can also act as an opportunistic pathogen in humans to cause a wide range of symptoms, including endophthalmitis, diarrhea, and fever [48,49]. In vitro work showed that a protein called AopP of *A. salmonicida* can physically interact with NF-κB proteins and prevents its translocation to the nucleus [50]. However, the significance of this effect in vivo was not appreciated until studies using fruit flies. *Drosophila* experiments showed AopP expression in hemocytes or imaginal disc epithelium led to a severe reduction in the levels of AMPs [51]. Both Drosomycin and Diptericins were affected, suggesting that both Toll and IMD pathways were simultaneously suppressed. Consistent with the findings from prior cellular studies, the authors found that ectopic expression of AopP suppresses nuclear translocation of both Relish and DIF (Figure 1). In addition, they found that AopP can also act in a pro-apoptotic manner in the *Drosophila* eye by inducing the activation of Caspase-3 to facilitate its cleavage (Figure 3). Flies expressing AopP in hemocytes also die quickly when challenged with a secondary infection with opportunistic gram-positive and gram-negative pathogens (*Micrococcus* and *Erwinia* species [52]), suggesting that this protein can disrupt the fly immune system through inhibition of NF-κB function.

### 2.3. HTLV-1: Tax1 in NF-κB (IMD) Signaling

Human T Cell Lymphotropic Virus type 1 (HTLV-1) is a retrovirus that places individuals at high risk for developing Adult T cell lymphoma (ATL) [53]. HTLV-1 infection initiates the onset of T cell lymphoproliferative malignancies that underlie ATL pathology. This is of particular concern as ATL has a poor prognosis [54]. The Tax1 transactivator of HTLV-1 mediates this damaging process, primarily through activating the NF-κB pathway [55]. Although insects lack a cell type that is directly homologous to lymphoid cells, including T cells, they are still very useful to understand the molecular functions of Tax1 based on conserved immune signaling pathways on which Tax1 acts. Previous work showed that Tax1 undergoes several posttranslational modifications, including SUMOylation and ubiquitination [56]. These modifications are considered to alter the activity, turnover, and subcellular localization of Tax1 [57]. Work in *Drosophila* combined with human cell-based assays identified a novel posttranslational modification on Tax1 and demonstrated the in vivo significance of this alteration [58]. In this study, Hleihel et al. first showed that Tax1 could undergo Urmylation, a process in which Urm1 (Ubiquitin-related modifier 1), a ubiquitin-like protein, is covalently conjugated to the target protein [59] (Figure 1). Using *Drosophila*, the authors then showed that over-expression of Urm1 changes the subcellular localization of Tax1 from the nucleus to the cytoplasm, leading to the activation of the IMD pathway increasing the expression of *Diptericins*. Based on these data, the authors proposed that Urmylation of Tax1 causes nuclear export of this protein, and this alteration in the subcellular localization facilitates the activation of the IMD branch of the NF-κB pathway by allowing Tax1 to interact with cytoplasmic proteins that regulate this pathway.

The effects of Tax1 expression in *Drosophila* host cells were further studied by assessing the effect of over-expressing this protein in the eye and in blood cells using *GMR (Glass Multiple Repeat)-GAL4* or *Pxn (Peroxidasin)-GAL4,* respectively [60]. Tax1 expression causes a disruption of the pattern of the compound eye when expressed in the developing eye imaginal disc. When Tax1 is expressed in plasmatocytes, fly immune cells with phagocytic activity, they undergo proliferation. These two phenotypes depend on *kenny*, the *Drosophila* homolog of IKKγ (encoded by *NEMO* in humans), which inhibits the function of IκB (Figure 1). Knockdown of *kenny* in Tax1 expressing cells using UAS/GAL4 driven RNAi can rescue the eye phenotypes. Interestingly, the expression of a Tax protein (Tax2) from a related virus, HTLV-2, does not show these same effects [61]. This is noteworthy because, unlike HTLV-1, HTLV-2 is not oncogenic [62]. Further work in cultured human T cells also demonstrated that Tax1, but not Tax2, induces expression of OX40L, a tumor necrosis factor ligand, by binding to an IKK [63]. This and additional studies corroborate to show NF-κB signaling becomes misregulated when Tax1 is over-expressed [58], demonstrating how data from *Drosophila* can facilitate the understanding of why related viruses can cause different diseases in humans.

### 2.4. HIV: Vpu in NF-κB (Toll) Signaling

Human Immunodeficiency Virus (HIV) is a retrovirus that infects B cells and subsequently leads to the development of acquired immunodeficiency syndrome (AIDS) [64]. The HIV genome contains nine genes that encode 15 proteins [65]. HIV Viral protein U (Vpu) is a viral accessory protein with a vast array of functions [66]. These include the downregulation of CD4 receptors in host cells, facilitation of viral release, modification to protein trafficking, and disruption of membrane integrity. Previous mammalian cell culture work also showed that Vpu inhibits the expression of pro-apoptotic genes downstream of NF-κB via phosphorylation-dependent interactions with β-TRcP (Slmb in flies), an E3 ubiquitin ligase that mediates the degradation of IκB [67,68]. Phosphorylated forms of Vpu do not have the ability to interact with β-TRcP or have marked effects on cultured human T cells [68]. Leulier et al. expressed Vpu in the fly fat body using the UAS/GAL4 system and showed it disrupts Toll signaling [69]. Interestingly, this manipulation weakened the fly’s ability to manage subsequent fungal infections, similar to the increased susceptibility to opportunistic fungal infections observed in AIDS patients [70]. Vpu specifically prevents the degradation of the IκB protein Cactus (Figure 1). Because Cactus degradation is required for the nuclear translocation of both DIF and Dorsal, Vpu inhibits the Toll signaling branch of the NF-κB pathway. Animals defective in Toll signaling cannot express *Drosomycin* in fat body cells, leading to immunodeficiency. Interestingly, activation of the IMD pathway and the expression of *Diptericins* are not affected by Vpu expression, indicating that Vpu specifically affects Toll signaling in this context. None of these effects were observed when the phosphomimetic form of Vpu that cannot bind to Slmb was expressed [69], in agreement with previous in vitro results. Recent work using cultured human cells demonstrated that Vpu-deficient HIV strains cause substantially stronger immune responses to viral infection when compared to wild-type strains, and this process was shown to be NF-κB dependent [71], corroborating the results obtained from *Drosophila*.

### 2.5. HIV: Nef in JNK Signaling

Nef (Negative Factor) of HIV is a protein produced in the early infection stage that is required for progression to AIDS [72]. Researchers became interested in Nef because several long-term HIV positive individuals who did not rapidly progress to AIDS were infected with HIV strains deficient in *Nef* [73]. Like Vpu, Nef has been implicated in immune suppression, primarily through downregulation of cell surface immunological proteins such as CD4 [74]. Previous work primarily using cultured cells showed that Nef undergoes a post-translational modification known as myristoylation, a covalent lipid-modification critical for its activity [75]. In vivo studies were performed using flies that expressed wild-type or myristoylation-defective Nef proteins under the control of the UAS/GAL4 system [76]. Expressing Nef in the developing *Drosophila* wing caused morphological defects, accompanied by an increase in Caspase activation and subsequent apoptosis. These phenotypes were not observed when mutant Nef was expressed, demonstrating that myristoylation is critical for Nef activity in vivo. The authors also observed a strong genetic interaction between Nef and components of JNK signaling. Co-expression of either *bsk* (JNK) or *hep* (JNKK) in conjunction with Nef enhanced the wing apoptosis phenotype while expressing Nef in heterozygous animals lacking one copy of either gene significantly suppressed this defect [76]. These data show that in addition to Vpu inhibiting Toll/NF-κB signaling, Nef likely potentiates JNK signaling upon HIV infection (Figure 2), suggesting that proteins encoded in the HIV genome affect the immune system in multiple ways. Further co-expression studies of Vpu and Nef may provide additional insights into how these signals influence one another to benefit the survival and spread of the virus within the human body.

## 3. Pathogenic Proteins that Affect Phagocytosis and Apoptosis

As discussed earlier, activation of the NF-κB pathway and production of AMPs is a critical line of defense against infection in *Drosophila*. This is, however, far from the only mechanism by which the host combats pathogens that are deleterious for its fitness and survival. Phagocytosis is a mechanism by which invading pathogens can be destroyed before they are able to release toxins or infect host cells [77]. In *Drosophila*, pathogen phagocytosis is primarily mediated by a subtype of hemocytes (blood cells) called plasmatocytes [78]. The entire process of phagocytosis requires complex cell dynamics, permitting alterations in motility, rapid engulfment, and effective degradation of the invading pathogen. Damage to protein machinery that modulate cell morphology or dynamics, such as cytoskeletal elements (e.g., actin filaments) and small GTPases (e.g., Rho), can severely limit or block phagocytosis. Some pathogens have developed methods to attack these processes to escape being engulfed by immune cells.

Apoptosis is another mechanism the host uses to control infectious pathogens [79]. When infection stress exceeds a certain level, cells lose the ability to maintain themselves and undergo programmed cell death, which can be beneficial for the organism as a whole. In *Drosophila*, pro-apoptotic genes *hid*, *grim*, and *reaper* play critical roles in apoptosis by inhibiting the activity of the anti-apoptotic protein DIAP1 (Death-associated Inhibitor of Apoptosis 1, also known as *thread*) [80,81]. DIAP1 functions to suppress the activation of Caspases, which are evolutionarily conserved executors of the apoptotic cell death program. Interestingly, some pathogens possess virulence factors, which seemingly have opposite effects on cell survival. Some factors facilitate cell death to damage specific host cells for their benefit, while others inhibit apoptosis in order to keep their host alive as long as possible to support pathogen proliferation and survival.

### 3.1. Pseudomonas aeruginosa: ExoS in Phagocytosis

*Pseudomonas aeruginosa* is a gram-negative bacterium that is an opportunistic pathogen that affects immunocompromised hosts, notably patients with severe burns and individuals with genetic conditions such as cystic fibrosis [82]. Colonization by *P. aeruginosa* can cause sepsis, during which the bacteria can form enduring biofilms that are difficult to eradicate with conventional antibacterial medications (see Section 5.2 as well). In addition, several strains of *P. aeruginosa*, especially those that acquire multidrug resistance, are of medical concern due to their role in causing ventilator-associated pneumonia [82]. *P. aeruginosa* is a unique pathogen in that it primarily disrupts host immunity, not at the levels of immune response signaling pathways, but at the level of phagocytosis. *P. aeruginosa* introduces Exoenzyme S (ExoS) into the host cell cytoplasm, which inhibits the host cells’ ability to engulf pathogens. ExoS targets multiple host proteins including small GTPases Rho, Rac, and Cdc42 to mediate its function [83]. Exoenzyme proteins are also crucial for disrupting the endothelial monolayer of vasculature, allowing the pathogen to invade additional body tissues and mediate disease progression [84]. Since ExoS expression was positively correlated with a worse prognosis [85], a fly model was developed to study the significance of this protein in vivo based on the UAS/GAL4 system [86]. Overall fly viability was unaffected by ubiquitous ExoS expression, but when ExoS-expressing flies were challenged with *P. aeruginosa*, their viability was significantly reduced. Interestingly, fat body specific expression of ExoS did not show this defect, and expression of AMPs was not altered upon this manipulation, suggesting the mechanism of action is NF-κB independent. In fact, expression of ExoS in hemocytes was sufficient to cause high susceptibility to *P. aeruginosa* infection, suggesting a role of ExoS in this cell type. The authors further showed that the N-terminal GAP (GTPase Activating Protein) domain of ExoS (ExoSGAP) alone is sufficient to perturb phagocytosis by interfering with the function of Rho family GTPases. In a subsequent paper, the same authors showed that Rac2 is the major Rho family GTPase that mediates the effect of ExoSGAP [86]. *Rac2* knockout flies phenocopied flies over-expressing ExoSGAP, including defects in phagocytosis, without alterations in NF-κB signaling. In addition, co-expression of Rac2 was able to neutralize the effect of ExoSGAP over-expression, further supporting the model that ExoSGAP inhibits phagocytosis by preventing proper Rac2 function [87]. This work shows that *Drosophila* can be used to study the role of pathogenic proteins that modulate the immune system in a way that is independent of classical immunosignal pathways, broadening the utility of *Drosophila* in infectious disease research.

### 3.2. SARS-CoV-1: 3a and M in Apoptosis

An outbreak of coronavirus between 2002–2003 in China caused severe respiratory symptoms in people who became infected, resulting in the death of 10% of infected patients [88]. This virus, which was named SARS-CoV-1 (Severe Acute Respiratory Syndrome Coronavirus, also known as SARS-CoV), is a relatively large (~30 kb) single-strand positive-stranded RNA virus that encodes a number of structural and nonstructural proteins [89,90]. To this day, many mysteries surround the exact pathogenesis of SARS. What we do know is that lung damage in SARS patients seems to correlate with high viral titer and increases in monocyte, macrophage, and neutrophil lung infiltration [91,92]. Viral infection also induces activation of several signaling cascades, including MAPK and PKB (Protein Kinase B)/AKT pathways as well as apoptotic pathways [93,94,95]. Several viral proteins have been implicated as mediators of these effects, including the SARS-CoV-1 Membrane (M) protein and the 3a protein. M is one of three structural proteins that make up the viral particle and also plays a central role in viral morphogenesis [96]. 3a is a nonstructural protein that does not have any significant homology to known protein families [97]. Although the precise function of 3a is still unknown, one study has shown that it possesses the capability to multimerize and function as an ion channel [90].

*Drosophila* biologists have studied these SARS-CoV-1 proteins using an over-expression-based strategy, which elucidated novel functions of these two proteins. Expressing M in the developing eye induced a rough eye phenotype, associated with increases in apoptotic activity in the developing eye imaginal disc [98]. By performing genetic modifier screens of this rough eye phenotype, the authors identified that over-expression of Pdk1 (Phosphoinositide-dependent kinase 1) could significantly suppress these rough eye and increased apoptosis phenotypes caused by M over-expression. Furthermore, the authors found that over-expression of M affected the phosphorylation status of fly Akt1, a core kinase in the PKB/AKT pathway that Pdk1 is known to act on. Based on these data, the authors argued that M induces apoptosis in cells by modulating the activation status of the Pdk1-Akt1 axis of the PKB/ATK pathway [98] (Figure 3). This finding was later confirmed in a mammalian system [99], showing the usefulness of unbiased screening approaches in *Drosophila* to identify unknown target genes and pathways of pathogenic factors.

Ectopic over-expression of 3a was also shown to cause an eye morphology defect similar to M [100]. The rough eye phenotype observed upon 3a expression is associated with increases in apoptotic cells and was suppressed with co-expression of an anti-apoptotic protein or a caspase inhibitor [100]. In a subsequent paper, the same group used this phenotype to perform structure-function analysis of the 3a protein and identified several amino acids that were important for its function [101]. Importantly, mutations that were previously shown to alter 3a’s ability to form an ion channel diminished the eye morphology and excessive apoptosis phenotypes, providing a molecular link between the channel activity identified from in vitro studies and severe in vivo phenotypes affecting cell survival. Although the precise mechanism by which 3a causes apoptosis is not known, these studies demonstrate the value of *Drosophila* in understanding the function of coronavirus proteins, paving the path to studying the role of SARS-CoV-2 proteins that are responsible for the COVID-19 pandemic. Indeed, a manuscript testing the role of 3a in SARS-CoV-2 using UAS/GAL4 system in vivo in flies was very recently uploaded onto a preprint server *bioRxiv* [102]. In this manuscript, the authors argue that similar to the 3a protein from SARS-CoV-1, 3a (referred to as ORF3a in the manuscript) from SARS-CoV-2 also induces apoptosis in the fly nervous system when over-expressed. They also show this protein activates both Toll and IMD pathways without affecting the JNK pathway. They further argue that “3a is the major virulence factor contributing to SARS-CoV-2 induced neurotropism” based on the phenotypes they observed upon overexpressing this protein in the nervous system [102]. Although this manuscript needs to undergo peer review, it is a prime example of how *Drosophila* researchers can quickly study an emerging virus and offer mechanistic insights that can be further tested in mammalian species. Furthermore, we foresee that by assessing genetic interactions between SARS-CoV-2 proteins with fly or human proteins, based on co-overexpression methodologies as well as RNAi and somatic CRISPR strategies, one will be able to understand which host-protein interactions, identified through human cell-based experiments, have relevance to COVID-19 pathogenesis [103,104].

### 3.3. HIV: Vpu in Apoptosis through JNK Signaling

As discussed earlier, experiments in *Drosophila* demonstrated that the HIV Vpu protein has the capacity to downregulate the Toll-branch of the NF-κB pathway in order to modulate the immune system [67,68,71]. In another study in flies, Vpu was also shown to increase the rate of apoptosis when expressed in the developing *Drosophila* wing [105]. Vpu causes morphological alterations to the wing when expressed in the developing animal using the UAS/GAL4 system. Interestingly, this effect was also observable, although with reduced expressivity, when a phosphomimetic form of Vpu was expressed. Because this form of Vpu lacks the ability to bind Slmb (β-TRcP) and is, therefore, unable to activate the Toll pathway [69], Vpu seems to be inducing apoptosis in a manner that is independent of NF-κB signaling. Overexpression of *DIAP1* strongly suppressed the wing phenotypes caused by Vpu over-expression, suggesting apoptosis is the main cause of the observed morphological defects. Downregulation of *grim*, *reaper*, and *hid* had similar effects as DIAP1 overexpression, strengthening the apoptotic hypothesis. Finally, through epistasis experiments, members of the JNK pathway upstream of *hep* (JNKK) were implicated as the targets of the Vpu-induced, caspase-dependent apoptotic cascade (Figure 2). Interestingly, the phosphomimetic form of Vpu was sufficient to induce JNK pathway activation in this context, similar to the wild-type protein. In summary, Vpu can trigger different signaling pathways in a phosphorylation-dependent (Toll activation) and independent (JNK activation) manner. This suggests that HIV, and likely other pathogens, may affect different signaling pathways depending on the state of the host cell to maximize its benefit.

### 3.4. EBV: BZLF1 and BRLF1 in Apoptosis and Cell Proliferation

Epstein Barr virus (EBV) is a member of the herpesvirus family that is widespread among the population [106]. Roughly 90% of adults become infected and acquire adaptive immunity at some point in their life. However, in some cases, EBV causes a wide range of diseases, including mononucleosis and increased susceptibility to several types of cancer. Current estimates predict EBV infection is directly responsible for nearly 150,000 cancer deaths per year worldwide [107].

One of the two early-immediate protein encoding genes, *BZLF1* (hereafter referred to as ‘*Z*’), is responsible for transitioning the virus from a latent stage to a lytic stage [108]. Z contains an AP-1 (Activator Protein-1)-like DNA binding domain, allowing it to bind to promoters within the viral genome [109]. Z also interacts with multiple transcription factors, including p53, CBP (CREB (cAMP Response Element-Binding Protein) Binding Protein), and NF-κB [110]. Although Z has been extensively studied through biochemical and cell biological assays, functional studies using in vivo models have lagged behind. Transgenic *Drosophila* expressing Z in the eye had marked disruption of eye morphology, including eye size reduction and ommatidial (units of eight photoreceptors and accessory cells that comprise the fly compound eye) disruptions [111]. This phenotype persisted even when a form of Z that cannot bind to DNA was expressed, suggesting this effect is mediated by protein-protein interactions rather than protein-DNA interactions. They also discovered that expression of Z both inhibited cell proliferation and increased apoptosis, in agreement with previous data based on in vitro mammalian cell culture experiments [112]. To identify the genes that work with Z in this context, the authors performed a genetic interaction screen. They identified *shaven,* the fly ortholog of the human paired family transcription factors *PAX2*, *PAX5,* and *PAX8*, as interactors (Figure 3). Biochemical work in human cells further determined that Z interacts with and inhibits PAX5 transactivation, suggesting these physical and genetic interactions are evolutionarily conserved between fly and mammals [113].

Similar to *Z* (*BZLF1)*, another early-immediate gene, *BRLF1* (hereafter referred to as ‘*R*’), of EBV has also been previously studied in the context of cell proliferation and survival. Unlike *Z*, which primarily inhibits cell cycle progression [112], R promotes cell cycle progression while, paradoxically, also promoting senescence in cell culture [114,115]. Previously, R was found to interact with and alter the function of several transcriptional regulators, including CBP, Rb (Retinoblastoma protein), and MCAF1 (MBD1-containing chromatin-associated factor 1) in mammalian cells [115,116]. The same group that investigated the function of *Z* in *Drosophila* studied the function of *R* in the fly eye [111]. These experiments showed that R promotes cell proliferation rather than inhibiting it like Z. This work also identified genetic interactors of *Z* and *R*, most of which are conserved between flies and humans. Of these host genes, three genes that are involved in apoptosis and cell growth (*p53*, *Tor* (*Target of rapamycin*), and *reaper*) modified the phenotypes caused by over-expression of both Z and R in opposite directions (Figure 3), suggesting that these two viral proteins may converge on common molecular and cellular pathways to fine tune cell survival and proliferation. Interestingly, co-expression of both Z and R in the fly eye can suppress each other’s phenotypes [111]. These results are intriguing, considering that the two proteins are expressed at the same time during EBV infection and are both required for the lytic activity of the virus [108]. Simultaneous expression of two antagonistic proteins may optimize the cellular environment for viral replication and survival by suppressing excessive proliferation or extensive cell death of the host cell.

## 4. Pathogenic Proteins that Affect Fundamental Cellular Processes and Developmental Signaling Pathways

So far, we have taken a close look at how certain host pathogenic factors interact with the immune system to overcome host defenses. Virulence factors can help bacteria and viruses evade host immunity through different mechanisms to facilitate infection. Some virulence factors can impact tissue and organ homeostasis after the host is infected, often hijacking endogenous cellular mechanisms for the benefit of the pathogen. Just as flies are useful for studying pathogen-immune interactions, they are also useful for studying how virulence factors interact with other fundamental cellular mechanisms in order to cause damage. In this section, we will discuss some pathogenic proteins from viruses and bacteria that impact fundamental cellular processes such as cell proliferation, cell adhesion, and cell polarity, as well as developmental signaling pathways including Notch, Hedgehog, RTK, JNK, and JAK-STAT.

### 4.1. Zika Virus: NS4A in Neural Stem Cell Survival and Proliferation

Zika virus is a mosquito-borne flavivirus closely related to Dengue and West Nile viruses. Zika was originally isolated in Uganda in the 1940s [117], but it was not until the twenty-first century that it became associated with severe neurological symptoms as the virus spread from Africa through Southeast Asia to the Pacific and finally to South America. During this time, Zika began to be associated with neurologic sequelae, including Guillain–Barre syndrome [118] and congenital microcephaly [119]. Several studies have been performed to understand how new strains of Zika virus cause strong neurodevelopmental phenotypes based on case studies and model systems [120,121]. One study took a comparative proteomics approach to identify host proteins that physically interact with proteins coded by the genome of two flaviviruses, Dengue and Zika [122]. The major goal of the study was to determine if viral factors can contribute to microcephaly by finding human proteins that strongly interacted with Zika but not or only weakly with Dengue, which has not been linked to microcephaly. From this work, biochemists found that a Zika virus protein, Nonstructural 4A (NS4A), specifically interacted with ANKLE2 (Ankyrin repeat and LEM domain containing 2) in a cultured human cell line. Previously, our group reported that rare variants in *ANKLE2* segregated in a family with severe congenital microcephaly and that a loss of function allele of *Drosophila Ankle2* also presented with a small brain volume phenotype [123], suggesting a link between *ANKLE2* and brain development in flies as well as in humans. Additional congenital microcephaly patients who carry deleterious variants in this gene were subsequently identified by several clinical genetic research groups [124,125], establishing *ANKLE2* as a bona fide microcephaly causing gene in humans.

To investigate whether the physical interaction between Zika NS4A and human ANKLE2 was meaningful, we ectopically expressed Zika NS4A in the developing third instar larva brains of *Drosophila* using the UAS/GAL4 system and found that this manipulation caused a reduction in brain lobe volume [122]. Interestingly, this defect could be rescued by co-expression of wild-type fly Ankle2 or reference (wild-type) human ANKLE2, suggesting that NS4A is acting through ANKLE2 to cause microcephaly in flies. To support this idea further, we found that expression of NS4A in a fly that is in a genetically sensitized background for *Ankle2* (heterozygous for a hypomorphic allele of *Ankle2*) caused a significantly stronger microcephaly phenotype compared to flies that express NS4A in a wild-type background, demonstrating that Zika *NS4A* and fly *Ankle2* genetically interact in vivo. In parallel to this work, we identified a genetic pathway in which *Ankle2* acts to regulate the asymmetric division of neural stem cells in the *Drosophila* brain [126]. In this model, *Ankle2* regulates the function of *ballchen* (*ball*, also known as *VRK1*), which in turn regulates the proper function of a group of genes that control cell polarity during asymmetric division of neuroblasts, including *l(2)gl* (*lethal (2) giant larvae*), *aPKC* (*atypical protein kinase C*), *bazooka* (also known as *par-3*) and *par-6*. These genes are important for the self-renewal of neural stem cells and the production of neurons. In *Ankle2* mutants, subcellular localization of key components of the asymmetric cell division pathway is disrupted. Strikingly, NS4A expression induced phenotypes similar to asymmetric localization that were also rescued by genetic modulation of downstream components in the *Ankle2* pathway. Since many genes in this pathway are conserved between human and flies, and several human genes have been linked to developmental brain disorders, including orthologs of *ball* (*VRK1* [127,128]), *l(2)gl* (*LLGL1* [129]), and *bazooka* (*PARD3B* [122]), NS4A likely contributes to Zika-mediated microcephaly by perturbing this evolutionarily conserved asymmetric cell division pathway in human (Figure 3). Although it is not clear whether any of the missense mutations in *NS4A* that have been acquired during Zika evolution contributes to the increased pathogenicity of strains found in South America [130], functional studies using *Drosophila* can be used to test such hypotheses. Furthermore, the fruit fly and the models developed above can be utilized to identify small molecules that suppress the small brain phenotypes in vivo. This has the potential to identify drugs that mitigate the effect of Zika-induced microcephaly. Although the best method to prevent this disorder is through avoiding mosquito bites and developing vaccines, such research avenues may lead to intervention strategies to rescue fetuses that have already been infected by the virus *in utero*.

### 4.2. Zika Virus: NS4A in JAK-STAT and Notch Signaling

In addition to its role in inhibiting ANKLE2 and modulating asymmetric cell division of neural stem cells, NS4A of Zika virus has been shown to modulate JAK-STAT and Notch signaling pathways in a fly model. To investigate host innate immune responses to Zika virus infection, several *Drosophila* models have been developed by exposing flies to Zika [131,132,133]. In one study, key inhibitors of the JAK-STAT pathway were found to be transcriptionally upregulated upon Zika virus infection [133]. JAK-STAT is an evolutionarily conserved signaling pathway that plays fundamental roles in development as well as in many pathophysiological contexts [134] (Figure 2). In *Drosophila*, this pathway can be activated by cytokines during inflammation and can further upregulate a number of downstream targets, including genes that are involved in humoral and cellular immune responses [135]. The authors further showed that JAK/STAT signaling activity was indeed reduced in flies infected by Zika infection and identified that NS4A has the ability to suppress JAK-STAT signaling activation (Figure 2). This conclusion was drawn from an experiment in which over-expression of NS4A in the fly eye was able to inhibit the tissue overgrowth phenotype caused by over-expression of an activated form of *hopscotch*, a JAK kinase [133]. In the same study, the authors also showed that NS4A could inhibit Notch signaling, a developmental pathway that has many roles in immunity and the nervous system [136,137,138] (Figure 4). Because the molecular mechanisms by which NS4A modulates these signaling pathways are still unknown, further mechanistic studies in flies and other systems are required. Together with the findings that Zika virus NS4A affects neural cell survival and asymmetric cell division [122,126], NS4A may have multiple targets that contribute to virus pathogenicity, providing an interesting target for drug development. Functional studies on other proteins and noncoding RNAs encoded in the Zika virus genome in flies will also be of great interest, especially since point mutations in some of these genes have been proposed to be associated with increased pathogenicity of the virus as it made its way to South America [139,140].

### 4.3. Bacillus Anthracis: LF and EF in Multiple Signaling Pathways and Cell Adhesion

*Bacillus anthracis* is a gram-positive bacterium that causes anthrax poisoning, a disease that occurs primarily in livestock but can also infect humans [141]. Due to its spore-forming ability and its toxicity, *B. anthracis* has been used as a biological weapon during warfare and as a bioterrorism agent [142]. Two bacterial proteins heavily involved in anthrax pathogenicity are Lethal Factor (LF) and Edema Factor (EF). LF is a metalloprotease that has multiple substrates, including a number of MAPKKs, whereas EF is an adenyl cyclase that can cause large, unregulated increases in intracellular cAMP levels [143]. These two toxins work in concert as anthrax lacking either factor shows markedly reduced virulence [144]. Although mechanisms of how *B. anthracis* infects the host have been well studied, the molecular mechanisms underlying the end-stage events of anthrax, including cellular processes that lead to widespread vascular leakage and shock, had not been adequately explored prior to work in fruit flies.

In a pioneering study, transgenic flies that express EF and LF were generated to investigate the mechanisms by which these factors cause severe symptoms seen in anthrax patients [145]. Ubiquitous expression of LF caused complete embryonic lethality, whereas LF driven in the developing dorsal thorax caused a severe “dorsal cleft” phenotype. This defect, in which the left and right thorax primordia fail to properly fuse in the midline, had been observed in mutants that affect the JNK signaling pathway, including *hep* (JNKK) mutants [146]. This phenotype caused by LF expression in the thorax can be suppressed by co-expressing a constitutively active Hep in the wing, suggesting that LF interacts with and prevents proper JNK signaling upstream of JNKK [147] (Figure 2). The authors also found that expressing LF in the developing wing pouch causes small, scooped wings with venation defects. This phenotype had been previously identified in mutants that are defective in another MAPK pathway that acts downstream of EGFR (Epidermal Growth Factor Receptor) and involves *Dsor1* (*Downstream of raf 1*), a MAPKK homologous to mammalian *MEK* genes in ERK signaling [148,149]. Expressing LF in a genetically sensitized background for *Dsor1* enhanced the wing phenotypes, demonstrating that LF likely acts to suppress the EGFR/MAPK pathway through Dsor1 inactivation (Figure 4). These in vivo findings have been validated by in vitro assays that demonstrated that LF could cleave multiple *Drosophila* MAPKKs, including Hep (JNKK) in vitro [147]. Therefore, the work in *Drosophila* validates in vitro data in mammalian cells that LF acts on MAPKKs [150] and has the potential to inhibit the activity of multiple MAPK signaling (JNK and ERK) pathways in vivo.

Similar to LF, *B. anthracis* EF causes multiple defects, including lethality, when ectopically over-expressed in *Drosophila* using various GAL4 drivers [147]. Particularly in the wing, LF causes wing venation phenotypes similar to those seen in hypomorphic alleles of *hedgehog* (*hh*) [151]. Considering that EF was proposed to act as an adenyl cyclase through in vitro assays [152], the authors hypothesized that this wing defect was caused by hyperactivation of PKA (Protein Kinase A), which is known to be under the regulation of cAMP. PKA negatively regulates Hedgehog signaling by phosphorylating Ci (Cubitus interruptus), the core transcriptional factor in the canonical Hedgehog pathway that is homologous to mammalian Gli proteins [153] (Figure 4). Indeed, wing phenotypes caused by EF could be made worse by reducing the level of *hh* in flies. Additionally, expression of EF was able to counteract the lethality caused by Pka-R1 (also known as PKAr), a regulatory subunit of PKA which causes ectopic activation of Hedgehog signaling through inhibition of PKA when over-expressed [154]. These data together strongly support the model that EF hyperactivates PKA, likely through dramatic increases in cAMP in vivo (Figure 4).

Experiments using *Drosophila* also provided the first insights into how LF and EF function in a cooperative manner. By further assessing the phenotypes induced by LF and EF in the developing fly wing, the same authors realized that the two proteins synergistically inhibit Notch signaling [155]. While individual expression of LF or EF alone had minimal impact on Notch activation, co-overexpression of the two proteins caused dramatic loss of Notch activation, leading to wing notching. By further studying the mechanism by which the co-overexpression of LF and EF induces this phenotype, the authors identified that altered trafficking of the ligand Delta is at the root of this defect. Rab11, a small GTPase, and its effector Sec15 are two host proteins that are required for proper subcellular localization of Delta that is critical for proper Notch signaling [155,156,157] (Figure 4). Over-expression of EF alters the subcellular localization of Rab11, whereas LF reduces the number of Sec15 positive vesicles, both of which cooperatively contribute to defective Dl trafficking and subsequent Notch signaling defects (Figure 4). Importantly, the authors showed that such synergistic inhibition of Rab11-Sec15 mediated vesicle trafficking event not only affects Notch signaling but also impacts cell-cell adhesion in endothelial cells by altering the subcellular localization of Cadherins that form adherens junctions (Figure 5). This was further shown to cause an increase in blood vessel permeability in mice, providing a molecular handle to begin to understand one of the key pathogenic symptoms of anthrax, disruption of endothelial barrier integrity, for the first time [155]. In a subsequent study, the mechanism by which EF disrupts the endosomal trafficking of Cadherins was determined by again combining *Drosophila* experiments with mammalian cell-based assays [158]. In addition to acting through PKA to suppress the interaction of Rab11 and its effectors, EF was also shown to activate Epac (Exchange protein directly activated by cAMP), a cAMP-dependent activator of Rap1, which is a small GTPase that inhibits the fusion of recycling endosomes with the plasma membrane [159,160] (Figure 5). These studies demonstrate how approaches combining in vivo experiments in *Drosophila* with in vitro and in vivo experiments in mammalian systems, including mice, can lead to a breakthrough in the field, permitting scientists to understand how multiple pathogenic proteins can work in concert.

### 4.4. Vibrio Cholerae: CtxA in Notch Signaling and Cell Adhesion

*Vibrio cholerae* is a gram-negative bacterium that causes cholera, an ancient disease that has been plaguing humans since antiquity [161]. Although this pathogen can typically be managed in the modern era with antibiotics and supportive therapy, outbreaks continue to be a problem in regions of the world with limited access to advanced medical care. Cholera typically presents with severe diarrhea, which can quickly dehydrate those suffering from this condition [162]. Diarrhea by *V. cholerae* is triggered by an increase of cAMP synthesis within intestinal epithelial cells, which in turn causes a large chloride efflux through the CFTR (Cystic fibrosis transmembrane conductance regulator) channel and alterations of osmotic pressure that pulls water out of cells into the digestive tract [163,164]. The virulence factor that mediates this effect is cholera toxin (Ctx), which is a multiprotein complex comprised of six protein subunits [165]. One subunit, referred to as the A subunit (CtxA), has enzymatic activity, whereas the remaining five subunits are referred to as B subunits (CtxB) that bind to cell surface receptors to permit the entry of CtxA into the host cell [166]. Upon entry, CtxA can ADP-ribosylate Gαs, a subunit of the stimulating G-protein complex. This post-translational modification activates adenyl cyclases expressed in the host cell, leading to a massive increase in cAMP production similar to effects caused by *B. anthracis* EF (although through a different mechanism in different cell types).

While a number of studies of *V. cholerae* and Ctx have been performed using rodent and cell culture models, *Drosophila* was utilized to specifically study the molecular function of CtxA using a transgenic approach. The same group that studied anthrax toxins in flies expressed CtxA in the developing fly wing and found a phenotype similar to defects observed when from *B. anthracis* LF and EF were co-expressed [164]. This defect was also caused by defective Notch activation due to defective Delta ligand trafficking (Figure 4). The wing phenotypes could be fully rescued upon co-expression of an active form of Notch or wild-type Rab11 but became significantly worse when a dominant-negative form of Rab11 was introduced. In addition, these phenotypes worsened when Gαs was co-overexpressed with CtxA but were ameliorated when an adenyl cyclase encoded by *rutabaga* was knocked down. Hence, similar to *B. anthracis* EF, *V. cholerae* CtxA is likely causing a dramatic increase in cAMP levels in vivo in *Drosophila*, which in turn inhibits Rab11 mediated vesicle trafficking (Figure 5). Furthermore, the authors showed that CtxA interferes with Notch signaling and disrupts cell-cell adhesion in two human intestinal epithelial cell lines (CACO-2 and T84 cells), demonstrating this effect is evolutionarily conserved.

To further determine whether cholera can be truly modeled in flies, the authors expressed CtxA in midgut epithelial cells using a gut specific GAL4 driver [164]. CtxA expressing flies exhibited defects in intestinal epithelial integrity. This was realized when the flies were fed blue-dyed food and underwent ‘smurfing’, a term used to describe animals turning blue as the contents of the gut leak into the body cavity [167]. Immunohistochemical examination revealed CtxA-expressing flies show defects in the midgut, including reduction of E-Cadherin at the adherens junctions (Figure 5). Chronic expression of CtxA in midgut cells leads not only to smurfing, but also to gradual wasting of the flies [164], similar to what had been observed when flies were directly infected with *V. cholerae* [168]. Importantly, these phenotypes can be rescued upon expression of Rab11, demonstrating that vesicular trafficking defects are at the root of pathogenesis. These findings were further corroborated with experiments performed in mice, which also identified adherens junction abnormalities in CtxA treated intestine [164], thus demonstrating a conserved effect of CtxA on intestinal integrity in vivo. Together with Zika virus NS4A work discussed earlier [125,126], this work not only showcases the ability of fly researchers to study molecular functions of a pathogenic factor using *Drosophila* as a ‘living test tube’, but also highlights that flies can seemingly recapitulate human phenotypes for some infectious diseases, functioning as ‘preclinical disease models’.

### 4.5. HCMV: Immediate-Early Genes in Cell Adhesion

Human cytomegalovirus (HCMV) is a herpesvirus that is typically asymptomatic but can occasionally cause lethal infections in children and immunocompromised individuals [169]. Additionally, when HCMV infects pregnant mothers, it can cross the placenta and cause a congenital viral infection that may result in severe neurological impairments and craniofacial dysmorphisms [170,171]. Historically, CMVs have been difficult to study in vertebrate models due to the high species-specificity [172]. Therefore, a system in which individual proteins could be studied in vivo was in high demand. Out of ~200 open reading frames (ORF) encoded in HCMV’s large viral genome, a set of genes that become expressed immediately after infection is referred to as ‘*immediate-early (IE)*’ genes [173]. IE genes are thought to act to make host cells more amenable to viral replication through multiple mechanisms [174,175].

To identify specific cellular processes affected by HCMV IE genes, one study generated flies that allow co-expression of two major IE proteins, IE72 (also known as IE1-72) and IE86 (also known as IE2-86), ubiquitously in a temporally controlled fashion using a heat shock promoter [176]. Expressing these IE proteins during embryogenesis caused lethality, accompanied by defects that suggested disruption of cell-cell adhesion. Upon ectopic expression of IE proteins during gastrulation, the localization of Armadillo (β-Catenin in mammals), an adaptor protein that links E-Cadherin to the actin cytoskeleton with α-Catenin [176,177,178], shifted from the membrane to the cytoplasm (Figure 5). Interestingly, subcellular localization of E-cadherin, apical protein complex comprised of Bazooka-Par6-aPKC, and subapical complex composed of Crumbs-Stardust-Patj were shown to be relatively intact, suggesting that the defect caused by ectopic expression of IE genes is more or less specific to Armadillo. Interestingly, HCMV had been shown to increase vascular permeability in endothelial cells by alterations in cell adhesion [179]. HCMV infection has also been shown to alter the expression of many kinases and phosphatases with the potential to post-translationally modify Armadillo/β-Catenin [176]. Hence, the *Drosophila* model developed in this study can be used to further dissect the molecular mechanisms by which IE proteins alters cell adhesion and other cellular processes relevant to HCMV infection.

### 4.6. Helicobacter Pylori: CagA in Multiple Signaling Pathways, Cytoskeletal Organization and Microbiome Composition

*Helicobacter pylori* is a gram-negative bacterium that is a common cause of peptic ulcers. Although half the world is infected with *H. pylori*, only a small fraction of people will ever present with symptoms [180]. In addition to developing benign ulcers, symptomatic patients have an increased risk of developing gastric tumors [181], making research on this bacterium relevant to cancer research. The main virulence factor of *H. pylori* is CagA (Cytotoxin-associated gene A). Upon entry into the host cell, CagA becomes phosphorylated by Src-family kinases and binds to SHP-2 (protein phosphatase encoded by the *PTPN11* gene in humans), activating signaling pathways downstream of RTKs [182]. In addition to Src kinases and SHP-2, in vitro experiments using cultured cells have shown that CagA can physically interact with other RTK pathway proteins such as c-Met (a RTK encoded by *MET*), CrkL (an adaptor protein encoded by *CRKL*), and Grb2 (an adaptor protein encoded by *GRB2*) [183,184]. Given these interactions and the strong activation of RTK signaling upon entry into host cells, CagA has been suspected of mimicking the activities of Gab (Grb2-associated binder) family proteins. These proteins function as endogenous scaffolding molecules for multiple RTK pathways to cause hyperplasia of intestinal cells, which can facilitate tumorigenesis [182,185].

While *H. pylori* CagA has been extensively studied biochemically and using in vitro cell culture systems, the in vivo significance of its involvement in RTK signaling was not explored prior to the following study in *Drosophila*. In addition to the fly wing, as discussed above, the fly eye is an excellent tissue to study the function of genes and proteins involved in RTK signaling because this pathway is used reiteratively during the development of the compound eye [186]. EGFR is an RTK required for cell proliferation in early eye development as well as for the specification of the many cell types comprising the ommatidia, including R1-R6 photoreceptor cells [187]. Sevenless, in contrast, is an RTK that is required only for the specification of R7 photoreceptor cells that is activated by Boss (Bride of Sevenless) [188]. Both EGFR and Sevenless act through Dos (Daughter of Sevenless), a Gab family protein [189] (Figure 4). Historically, sophisticated genetic technologies have been used to genetically assemble the RTK pathway through studies of EGFR, Sevenless, and other components of this pathway in the fly eye [190,191].

To understand the role of CagA in RTK signaling, one group generated transgenic flies that allow the expression of wild-type CagA or a CagA lacking four of its EPIYA (Glutamate-Proline-Isoleucine-Tyrosine-Alanine) phosphorylation motifs. Expression of wild-type CagA caused dose-dependent eye morphology defects (rough eyes), while the CagA lacking EPIYA motifs did not show this phenotype, suggesting the importance of phosphorylation sites on CagA function. Furthermore, they found that expression of wild-type CagA can partially rescue the lethality of *dos* deficient flies, allowing them to survive into pupation, providing in vivo evidence that CagA can function as a Gab [192] (Figure 4). The ability of CagA to substitute for the lack of Dos was also tested in the eye by generating mosaic animals in which *dos* mutant clones generated using the FLP/FRT system [193] and expressing CagA using the UAS/GAL4 system in this tissue. From this assay, the authors showed that *dos* mutant cells expressing CagA were larger and more frequently found compared to *dos* mutant cells that lacked this protein, demonstrating that CagA can promote cell survival and proliferation in cells that lack *dos*. Importantly, the authors also showed that fly SHP-2 was epistatic to CagA by showing that the phenotypes caused by over-expression of CagA can be suppressed in a *corkscrew* (fly SHP-2 encoding gene) mutant background, demonstrating that the mechanism by which CagA affects RTK signaling discovered in mammalian cell based studies are conserved in *Drosophila* [192] (Figure 4).

To further understand how CagA causes a rough eye phenotype, the same group studied the relationship between the cytoskeleton and CagA [194]. Prior studies in cultured cells provided multiple indications that CagA regulates cell morphology by altering actin-based cytoskeletal networks [195,196,197]. Using the same transgenic flies, the authors found that expression of CagA in the developing eye disrupted epithelial integrity, causing ectopic furrowing of epithelial cells in the eye imaginal disc accompanied by abnormal F-Actin patterns. Interestingly, this phenotype was dependent on EPIYA motifs but independent of SHP-2, suggesting that a protein or proteins other than SHP-2 are responsible for the cytoskeletal phenotype. Since activation of non-muscle Myosin II complex was shown to regulate epithelial architecture in the developing eye [198], the authors focused on Myosin light chain (MLC, encoded by the *spaghetti squash* gene), a regulatory subunit of the complex that is regulated by multiple kinases [199]. In CagA expressing cells, MLC was enriched in ectopic furrows, suggesting it may be misregulated. Importantly, co-expression of a dominant-negative form of MLC was sufficient to suppress epithelial phenotypes caused by CagA expression, suggesting that CagA may cause over-activation of MLC in vivo. The authors also showed over-expression of a constitutively active Rho GTPase, a positive regulator of MLC activity [200], can phenocopy the effect of CagA in vivo. Using a cell-based assay in *Drosophila*, the authors further demonstrated that CagA could regulate the subcellular localization of MLC in a Rho-dependent manner, leading to their model that CagA modifies actin cytoskeleton through activation of MLC via Rho (Figure 5). This study provided important insights that proteins other than SHP-2 can mediate the pathogenic effect of CagA, underscoring the value of focusing on morphological phenotypes elicited by ectopic expression of pathogenic proteins in seemingly unrelated cell types (i.e., expressing a protein from bacteria that infects human intestinal cells in the eye precursor cells in *Drosophila*).

In addition to the eye, effects of CagA expression have been studied in fly wings and revealed CagA affects JNK signaling [201] (Figure 2). Ectopic expression of CagA in the epithelial cells of the developing wing induced apoptotic clusters in a dose-dependent fashion, leading to a severe decrease in wing size. This defect was notable because of its similarity to phenotypes observed from localized, but not ubiquitous, activation of the JNK pathway within the wing [202]. Indeed, CagA expression led to the activation of this pathway as judged by increased phosphorylation of Bsk (JNK) as well as expression of a reporter transgene (*puckered-lacZ*). The authors were also able to suppress apoptosis caused by CagA expression by co-overexpressing a dominant-negative form of Bsk. Moreover, the phenotype became stronger when Bsk and CagA were co-overexpressed, suggesting a synergistic relationship between the two proteins. One way JNK signaling in *Drosophila* can be activated by Eiger, a TNF (Tumor Necrosis Factor) superfamily ligand [203]. The authors unexpectedly found that Eiger is required non-cell autonomously in neighboring cells to facilitate phagocytosis of CagA expressing cells undergoing apoptosis. This finding suggested that cells undergoing apoptosis were actively being removed from the wing epithelium and not simply being lost during development. Also, this data indicated that JNK signaling acts both cell autonomously and non-cell autonomously in these processes. This study provided important insights into the complicated host inter-cellular interactions that can occur during infection.

Since JNK signaling can cause apoptosis or stimulate growth, depending on the activation status of the Ras oncogene [204], the authors investigated if CagA can genetically interact with a Ras allele that carries an oncogenic constitutive active variant (p.G12V, also known as Ras^V12^) in the same study [201]. Expressing Ras^V12^ alone in the fly eye causes overgrowth and tumor formation, though these tumors are not typically invasive [205]. Co-expressing CagA with Ras^V12^ leads to a tumor that can spread into the fly central nervous system, demonstrating that CagA activation of the JNK pathway can synergize with the oncogenic effect of hyperactive Ras signaling. This study also revealed that some genes involved in cell polarity genetically interact with CagA to modulate the apoptotic phenotype in the wing. Knockdown of *dlg1* and *l(2)gl*, which encode proteins that are enriched in the basolateral membrane of epithelial cells, significantly enhanced the apoptotic defect seen upon CagA overexpression [201]. Since these genes are considered neoplastic tumor suppressor genes, a group of genes that cause tissue overgrowth when lost in *Drosophila*, CagA may also interact with mammalian tumor suppressor genes to mediate the oncogenic effect of *H. pylori*. Together with the finding that CagA genetically interacts with Ras, this work provides an excellent framework and model system to study how virulence factors, host oncogenes, and tumor suppressors work in concert to transform a benign ulcer into a life-threatening cancerous lesion in humans. When combined with further genetic interaction screens to identify enhancers and suppressors of phenotypes caused by CagA expression [206], *Drosophila* researchers can provide a list of potential pharmacological targets to combat the oncogenic effect of *H. pylori*

Finally, work in flies demonstrated CagA expression in the digestive tract could act on other bacteria in the gut to indirectly impact the host. *H. pylori* is known to affect the microbiome of its hosts, a key part of gastric and organism health [207]. Although this effect is likely to be mediated through interactions of many genes and proteins expressed by *H pylori*, other microbes, and host cells, studies in *Drosophila* suggested that CagA alone is sufficient to impact organism-level physiology via ROS (reactive oxygen species) production by other bacteria [208]. Overexpressing CagA with an intestinal stem cell-specific GAL4 driver led to an increase in cell proliferation and an upregulation of immune response markers such as *Diptericin* in the fly midgut. These are notable phenotypes as inflammation, and cell proliferation are key components of *H. pylori* pathogenesis in humans. CagA expressing flies reared in a germ-free environment did not have the same inflammation and proliferation defects as those raised in a conventional environment, suggesting the effect of CagA depends on the host microbiome. Indeed, CagA expressing flies showed dysbiosis of their gastric microbiota and showed colonization of *Lactobacillus brevis* that was not seen in wild-type control flies. *Lactobacillus* species including *L. brevis* and *L. plantarum* were identified as possible commensals that aid in cell-over proliferation phenotypes via stimulation of ROS in the midgut epithelium through uracil secretion [208]. Similar to previous studies in flies and other systems, this process was also shown to be dependent on the phosphorylation status of CagA because CagA lacking EPIYA phosphorylation motifs did not have the same effect as the wild-type protein. The importance of these EPIYA motifs continues to be appreciated as new strains of *H. pylori* are identified with varying EPIYA sequences associated with different cancer risks [209]. This makes *Drosophila* a potential in vivo tool to dissect these variant forms of CagA and how they affect their hosts.

### 4.7. HPV: E6 in Cell Polarity and Epithelial-to-Mesenchymal Transition

Human papillomavirus (HPV) is a non-enveloped DNA virus in the *Papillomaviridae* family that causes a sexually transmitted disease, although it can also spread dermatologically [210]. It is estimated that half of all humans will at one point be infected, but the infection typically resolves without any intervention if proper immune responses occur [211]. While the disease typically manifests as anogenital warts, HPV can also cause serious conditions such as cervical cancer, a major cause of death for women worldwide [212]. Two HPV proteins, E6 and E7, have been identified as cooperative oncoproteins implicated in progression from localized to metastatic HPV-induced cancers [213]. While E7 binds to the Retinoblastoma (Rb) protein and inhibits its ability to sequester E2F family transcription factors that regulate cell cycle progression [214,215], E6 has been shown to inactivate a number of tumor suppressors, including p53 as well as multiple cell polarity regulators such as DLG1 (scaffold protein), SCRIB (scaffold protein) and MAGI1 (membrane associated protein kinase) [216,217,218,219]. E6 acts by binding to its substrates and directing them towards ubiquitin-mediated proteasomal degradation through the recruitment of the E3 ubiquitin ligase, UBE3A (also known as E6 Adaptor Protein (E6-AP)) expressed by the host cell [220].

Transgenic flies capable of co-expressing HPV E6 and human UBE3A were developed to understand how E6 functions in vivo, and to further identify its targets [221]. While expressing E6 or UBE3A alone in the eye or wing did not cause any morphological defects, co-overexpression of the viral (E6) and host (UBE3A) proteins caused rough eyes and blistered/melanized wings. This reinforces the notion that biochemical interactions between these proteins are biologically significant and that E6 recruits UBE3A to execute its function. Further assessment of the cellular consequence of co-overexpressing E6 and human UBE3A identified disruption of cell adhesion and polarity in the eye as well as excessive apoptosis in the wing primordium. The authors attempted to determine which proteins identified previously in mammalian cell culture studies were the target of E6-UBE3A complex in vivo by assessing the expression of the fly orthologs of p53, DLG1 (Dlg1), SCRIB (Scrib), and MAGI1 (Magi). Experiments in the wing imaginal disc showed that while Dlg1, Scrib, and Magi can be degraded by co-expression of E6 and human UBE3A (Figure 5), p53 levels and subcellular localization does not change. The authors also showed that developmental defects in the wing seen upon E6 and human UBE3A co-expression could be partially suppressed by co-overexpression of Magi, demonstrating Magi loss is at least partially responsible for wing phenotypes. In addition, by performing a targeted genetic interaction screen focusing on signaling pathway genes, they found co-overexpression of E6 and human UBE3A alongside a dominant-negative form of Insulin receptor significantly worsened the eye phenotype caused by either E6 or human UBE3A alone, suggesting a synergistic interaction. Finally, the authors showed that E6 and UBE3A synergize with the oncogenic forms of Ras or Notch to cause epithelial-to-mesenchymal transition (EMT), a fundamental process for cancer metastasis [222]. These experiments laid the foundation to dissect p53-independent mechanisms of E6 tumorigenesis and provide a *Drosophila* model to study the molecular mechanism of EMT.

### 4.8. HIV: Tat in Cytoskeleton Organization and Protein Translation

HIV, which primarily infects the immune system, has also been shown to infect cells in the nervous system, leading to neurocognitive symptoms [223]. Tat (Trans-activator of transcription) is one of two regulatory proteins encoded in the HIV genome and is primarily known for its role in activating viral gene transcription [224]. Previous biochemical and cell biological work demonstrated Tat’s ability to interact with the host cell cytoskeleton [225,226]. Such interactions have been shown to cause wide-ranging defects during angiogenesis, apoptosis, and cell proliferation. Some have postulated that Tat may play a role in neurodegeneration seen in some HIV patients by altering the function of tubulin and actin-based cytoskeletal networks in neurons [227]. In order to better understand host-virus interactions occurring at the molecular level in vivo, one study over-expressed HIV Tat under the control of a heat shock promoter in the female germline [228], a fly organ in which the cytoskeleton has been extensively investigated [229,230]. One obvious phenotype noticed upon Tat over-expression during oogenesis was morphological abnormalities of dorsal appendages, structures that serve as gas exchange apparatus for the developing embryo [231]. This phenotype was likely caused by defects in dorso-ventral patterning of the oocyte, a process that depends heavily on Tubulin polymerization [232,233]. Indeed, the authors identified a defect in cytoplasmic streaming, a microtubule-dependent process in which mRNAs and proteins are delivered to the correct location within the oocyte [234]. Co-immunoprecipitation and co-immunostaining experiments demonstrated Tat and Tubulin physically interact in vivo. Additional in vitro experiments demonstrated that Tat decreases the polymerization rate of Tubulin. In a subsequent study, the same group performed microinjection of recombinant Tat into *Drosophila* syncytial embryos. Using this non-genetic method, the authors also showed that Tat impacts microtubule dynamics during mitosis and noted chromosomal segregation defects [235].

In addition to its role in affecting the proper organization of the cytoskeleton and impacting viral gene transcription, Tat has also been shown to affect key factors involved in translation based on *Drosophila* studies. By assessing the subcellular localization of ectopically expressed HIV Tat in the female germline, this viral protein was found to localize to the nucleolus of nurse cells, a group of polypoid cells with large nuclei that support the oocyte during development [235]. Nucleolar localization of Tat has been reported in mammalian publications [236], but this was primarily studied in the context of viral gene transcription. Using *Drosophila*, the authors showed that Tat interferes with pre-rRNA processing of ribosome biosynthesis by physically interacting with the ETS-18S region of pre-rRNA. Additionally, Tat can also bind to Fibrillarin, an essential component of the small nuclear ribonucleoprotein (snRNP) critical for pre-rRNA maturation [236]. This result was similar to what was previously shown in mammalian experiments [237]. The authors concluded these molecular interactions, Tat-pre-rRNA and Tat-Fibrillarin, both contribute to the decrease in 80S ribosome levels seen upon Tat over-expression [238]. These studies together demonstrated Tat impacts microtubule dynamics and protein translation. Although the precise molecular mechanism of HIV-associated neurocognitive disorders is yet unknown [239], cytoskeleton and translation regulation play critical roles in neuronal function and maintenance [240,241]. Studies using *Drosophila* that over-express Tat and other HIV proteins in the mature nervous system may provide hints to solve this mystery.

### 4.9. SV40: Large and Small T Antigens in Mitosis

Simian vacuolating virus 40 (SV40) is a non-enveloped DNA virus in the *Polyomaviridae* family that is primarily found in rhesus monkeys but can infect other mammalian species, including humans [242]. SV40 spread within the human population in the 1950s and 1960s because a significant fraction of polio vaccines administrated during this period were made using monkey-derived cell lines contaminated with the virus [243]. SV40 is a public health concern as it can cause tumors in some animals, such as hamsters [244]. Although SV40 has been extensively studied as an oncovirus by many scientists, whether SV40 is capable of causing cancer in humans is controversial and remains under debate [245,246].

The study of oncogenic viral proteins (oncoproteins) has advanced our understanding of how viral proteins interact with host proteins to modulate their functions. SV40 expresses two major oncoproteins: the large tumor antigen (LT) and the small tumor antigen (ST) [247]. Activities of both ST and LT are required to efficiently transform naïve human cells into immortal cells [248]. This suggests that the two proteins act on different targets expressed in the host cell. Through previous cell-based experiments as well as in vivo work in mice, LT was shown to act on several critical tumor suppressors and oncoproteins, including p53, Rb, and Myc [249]. ST instead interferes with Protein Phosphatase 2A (PP2A), an evolutionarily conserved protein phosphatase complex that regulates proteins involved in various steps of tumorigenesis [250,251].

Although the molecular functions of these viral oncoproteins from SV40 have been extensively studied in cell-based assays, and a number of rodent models have been generated to understand their roles in tumorigenesis in vivo, one study developed a fly model to specifically study how LT and ST can interfere with development [252]. In this work, the authors established flies in which the genomic region of SV40, which includes both LT and ST (early region), was placed under the control of UAS. Because LT and ST become expressed through alternative splicing of a single early region transcript, whether LT or ST or both proteins will be expressed upon GAL4-dependent transcription induction depends on its cellular context. Ectopic expression of the SV40 early region was shown to cause lethality in developing fly embryos due to mitotic spindle abnormality accompanied by the formation of supernumerary centrosomes that ultimately led to cell cycle breakdown. In these embryos, ST but not LT was expressed, suggesting that ST was likely to be the cause of chromosomal defects. Indeed, overexpression of ST alone phenocopied the defect caused by the SV40 early region transgene. The authors also showed ST that cannot bind to PP2A lacked this activity, and the effect of ST over-expression can be enhanced in a genetic background that is sensitized to PP2A. Finally, the authors identified proteins that act downstream of PP2A to mediate the mitotic defects caused by ST. One such protein was Cyclin E, a cell cycle regulator that acts with Cdk2 and is required for centrosome duplication. Overexpression of ST caused an approximately two-fold increase of Cyclin E in the fly embryo, suggesting that ST can transcriptionally or post-transcriptionally regulate the level of this host protein in vivo (Figure 3). Interestingly, independent studies published around the same time as this fly manuscript also suggested SV40 ST and Cyclin E act cooperatively to mediate cellular transformation [253,254], suggesting that these two proteins may function together to cause tumorigenesis in vivo. This study again highlights the value of *Drosophila* in uncovering evolutionarily conserved host proteins that mediate the function of virulence factors.

## 5. *Drosophila* Studies to Identify New Therapeutic Targets to Combat Infectious Diseases

As discussed in previous sections, flies have repeatedly proven to be excellent models to study human pathogens, thanks to the remarkable conservation of innate immune mechanisms, developmental signaling pathways, and fundamental cellular processes. The functions and effectors of individual virulence factors can be explored in vivo with relative ease in *Drosophila*, thanks to the rich genetic resources that are available to researchers in the field [5,255,256]. In this section, our discussion will focus on an exciting new frontier; utilizing *Drosophila* as a drug discovery platform against infectious diseases.

*Drosophila* can help identify key host proteins targeted by pathogenic factors, like *ANKLE2* that is inhibited by the Zika virus protein NS4A [122,125]. We have also seen *Drosophila* as a valuable tool to tease out subtle differences in homologous proteins encoded by pathogenic and nonpathogenic strains of similar viruses as in the case of HTLV Tax proteins [60]. Utilizing such knowledge and the reagents generated for fundamental biological studies, one can move ahead and use *Drosophila* as a tool not only to study virulence factor action but also to develop strategies to reduce the activity of these virulence factors in vivo. This is particularly important for pathogens that develop drug resistance and become difficult to combat using standard approaches. In this section, we will discuss two cases in particular. The first is a study that took an over-expression-based approach to study an influenza virus virulence factor M2, whose evolution has been implicated in drug-resistance. Through this study, the authors identified a V-ATPase inhibitor as a potential drug candidate [257]. The second study took an infection-based approach to investigate the function of enzymes that contribute to biofilm formation in *Pseudomonas* and how they affect the bacteria’s sensitivity to certain types of drugs treatments. This work revealed that two genes, *gshA* and *gshB*, involved in glutathione synthesis change bacterial sensitivity to ROS (reactive oxygen species), revealing a possible therapeutic avenue [258].

### 5.1. Influenza Virus: M2 in pH Regulation

The Influenza virus is a single-strand RNA virus in the *Orthomyxoviridae* family that causes the flu in humans and other species [259]. The Influenza virus is a constant threat to public health and safety because it can easily spread between hosts, including occasional interspecies spread. Because their genomes are highly mutagenic, people need to receive new rounds of flu shots every year to prevent contracting viral infections each flu season. Two proteins, Hemagglutinin (HA) and Matrix protein 2 (M2), play critical roles in the survival and spread of the Influenza virus. HA is an attachment protein that binds sialic acids residing on the surface of host cells and also functions as a membrane fusion protein to mediate viral entry into host cells. M2 is an ion channel required to lower the internal pH of the virion, which facilitates viral uncoating [260]. These proteins have been common targets of antiviral drug development given their necessity to the viral life cycle. Currently, several anti-influenza drugs are clinically used that target M2, including amantadine (e.g., Gocovri^®^) and rimantadine (e.g., Flumadine^®^). However, problematically new strains of influenza are becoming resistant to these drugs due to missense mutations that occur on M2 during viral evolution [257].

To generate a model of amantadine resistance and to potentially serve as a possible drug discovery platform, one group generated a transgenic fly strain that expresses Influenza virus M2 protein under the control of the UAS/GAL4 system [261]. As expected, based on previous biochemical studies, in vivo experiments showed that M2 acts as a functional proton channel, which was functional when ectopically expressed in flies. Expression of M2 in the eye and wing caused dose-dependent morphological phenotypes, and the authors further showed that M2 becomes localized to the plasma membrane and other intracellular membranes when expressed in a fly cell, similar to earlier reports in mammalian cells [262]. M2 overexpression was also shown to increase intracellular pH in *Drosophila*, consistent with earlier data from cultured cell-based experiments. Treating flies with amantadine was able to suppress the rough eye phenotype caused by ectopic M2 overexpression, providing evidence that this system can be used to identify novel genetic interactors or small molecules that can inhibit M2 activity in vivo. As a proof of concept, the authors performed a candidate-based screen by overexpressing M2 in *Drosophila* genetic backgrounds that were sensitized with mutations in genes involved in ion transport or pH homeostasis. Through this approach, they identified multiple genes that encode subunits of the V-ATPase, a vacuolar ATPase that is responsible for the acidification of endolysosomal organelles [263]. The authors further showed that Bafilomycin, a chemical inhibitor of V-ATPase, could decrease Influenza virus infection in a mammalian cell-based assay, whereas overexpression of V-ATPase subunits in the same model increases viral infectivity. Although Bafilomycin is unlikely to be a good antiviral drug to combat the flu due to its pleiotropic effect and high toxicity [264], this study provides a framework to utilize morphological phenotypes induced by overexpression of virulence factors in *Drosophila* as a functional readout to perform genetic and pharmacological screens.

### 5.2. Pseudomonas aeruginosa. gshA and gshB in Bacterial Stress Resistance and Biofilm Production

As discussed in a prior section, *P. aeruginosa* can cause nosocomial infections that subvert host immune responses. *P. aeruginosa* accomplishes this by expressing ExoS, which disrupts the hosts’ ability to phagocytose the pathogen. Infection often results in biofilm formation, which damages the host respiratory epithelium [82]. It has been noted that different strains of *P. aeruginosa* can have very different outcomes in infected patients. One group utilized flies to study the molecular mechanisms that relate to biofilm formation to determine what could account for these differences in disease outcome [258]. *gshA* and *gshB* encode two enzymes involved in glutathione synthesis. Glutathione is a small molecule that plays numerous roles in bacterial survival by maintaining cellular homeostasis. In addition to regulating biofilm production, glutathione also serves as an abundant antioxidant molecule to protect against oxidative stress [265]. The predominant PAO1 strain of *P. aeruginosa* contains both genes, which become upregulated upon exposure to diverse stressors, including ROS.

Infection-based assays in *Drosophila* demonstrated that *P. aeruginosa* strains lacking either *gshA* or *gshB* were less virulent than their wild-type counterparts [258]. Strains lacking both *gshA* and *gshB* were even less virulent, indicating an additive effect. Loss of *gshA* and *gshB* genes increased the sensitivity of *P. aeruginosa* to ROS and electrophilic stressors and decreased the motility of the bacteria, which likely makes these bacteria more susceptible to agents that act on these processes. These same manipulations also increased the formation of biofilms, which contribute to the pathogenicity of the bacteria and antibiotic resistance [266,267]. Although this study did not directly use *Drosophila* as a drug discovery tool, infection based virulence assays in *Drosophila* in vivo combined with biochemical and microbiological assays in vitro can reveal the Achilles’ heel of different bacterial strains to design effective targeted therapies. In addition, it is worth further mentioning that *Drosophila* is being used as a model to develop phage therapy against *P. aeruginosa* [268,269,270], a biological strategy that takes advantage of bacteriophages to combat antibiotic strains [271]. Hence, utilization of *Drosophila* beyond pharmacologic and genetic screening may also facilitate the development of novel types of therapies against bacterial and viral strains that are highly mutagenic.

## 6. Conclusions and Future Directions

As discussed throughout this manuscript, *Drosophila melanogaster* is a powerful system to study pathogenic mechanisms of virulence factors that affect humans as well as other species. Although typically thought of as a model organism best suited to study basic genetic and biological questions, flies are gaining more popularity as a powerful tool to study mechanisms underlying human genetic disorders [10,11]. The studies discussed here showcase how flies can be used to test a specific hypothesis based on data from in vitro experiments or to perform unbiased screens to identify novel downstream target genes and proteins of a specific pathogenic factor in vivo to develop novel hypotheses. The molecular mechanisms controlling innate immunity, developmental signaling pathways, and other fundamental cellular and biochemical pathways are well conserved in *Drosophila*. This, combined with powerful genetic techniques that can be used to perform probing experiments in a cost and time efficient manner, make flies a powerful model system. As such, fly biologists have made a number of contributions to understanding the molecular functions of proteins produced by viruses and bacteria.

In addition to challenging flies with certain pathogens that infect insect species, such as vector borne viruses and certain strains of bacteria and fungi, heterologous protein expression systems based on the UAS/GAL4 system provide a powerful complementary approach to dissect out the function of pathogenic proteins, one factor at a time. Expressing a single protein or multiple proteins from pathogens can give insights into disease mechanisms by determining on which host proteins they act. These systems can also be leveraged to study similarities and differences between closely related viral and bacterial strains. By expressing different variant forms of pathogenic proteins, one should be able to quickly and efficiently understand the evolution of new and evolving pathogens, including the Zika virus and SARS-CoV-2, both of which are crucial health concerns at the time of this publication. The availability of genetic reagents in the fly community also allows rapid follow-up work to be done by experimenting on how different genetic backgrounds, alleles, or knockdowns affect phenotypes induced by a specific pathogenic protein. These factors also contribute to making fruit flies an attractive drug discovery platform for translational research.

It is important to note, however, that there are certain limitations for *Drosophila* in immunological and infectious disease research. Flies rely mostly on innate immunity and lack cells that are critical regulators of the adaptive immune system, such as lymphocytes, including B cells and T cells. In addition, flies lack certain key branches of immune signaling, such as interferon responses and a genetically conserved complement system seen in mammals [272,273]. They also have unique hematological responses to certain pathogens, including crystal cell melanization and lamellocyte encapsulation responses that are more or less insect-specific [274]. These factors must be taken into account when determining if *Drosophila* is a suitable model for one’s disease, or disease process, of interest. One should always make sure that the mechanism of action for a pathogen of interest acts on a well-conserved pathway by attempting to validate the initial findings in *Drosophila* using in vitro and/or in vivo mammalian experimental systems. Rather than considering fruit flies as an ‘alternative model’ to study infectious diseases, we argue that one should think of this model organism as a ‘synergetic model’ that can boost the speed of discovery when combined with experiments performed in mammalian models, especially in vivo mouse models and human cell and organoid culture systems.

Understanding how pathogens evade host immunity is key to our understanding of infections. This is especially true for understanding how one infection can lead to subsequent infections, as seen in immune-compromised patients. Upon entry into the host, pathogens hijack host processes to facilitate their survival and proliferation. Processes such as cell division, cell adhesion, and developmental signaling pathways are impacted by infection, and many of these processes are evolutionarily conserved and have been extensively studied in *Drosophila*. Therefore, expressing pathogenic proteins in a context that may initially seem unrelated to human disease often provides a fundamental understanding of what virulence factors are doing in vivo in another context. Genetic assays performed by screening for morphological phenotypes caused by ectopic over-expression of pathogenic proteins are robust and often help identify new proteins that are inhibited or activated by the pathogen. Because these experiments, and flies as a model system in general, are also quite scalable, *Drosophila* can be used to assess the impact of novel small molecules, as well as extant drugs, to identify lead compounds to combat disease. We predict the number of researchers that utilize *Drosophila* for infectious disease research will increase over time, further enriching the repertoire of genetic tools available to the fly community to tackle important questions in this field.

In summary, experiments that have been performed in *Drosophila* over the past century have advanced nearly every aspect of our understanding of biology. Flies have been repeatedly shown to be useful in studying an array of topics, from basic genetic concepts to modeling rare human genetic diseases. The most crucial parts of cellular function are often the root cause of disease, both infectious and otherwise. These crucial parts have been well conserved through hundreds of millions of years of evolution to be studied in an array of model systems. Powerful, proven, and constantly evolving model organisms like *Drosophila melanogaster* should be considered a key part of biomedical research that can be used in a complementary fashion together with in vitro assays and mammalian experiments in vivo.

## Figures and Tables

**Figure 1 ijms-22-02724-f001:**
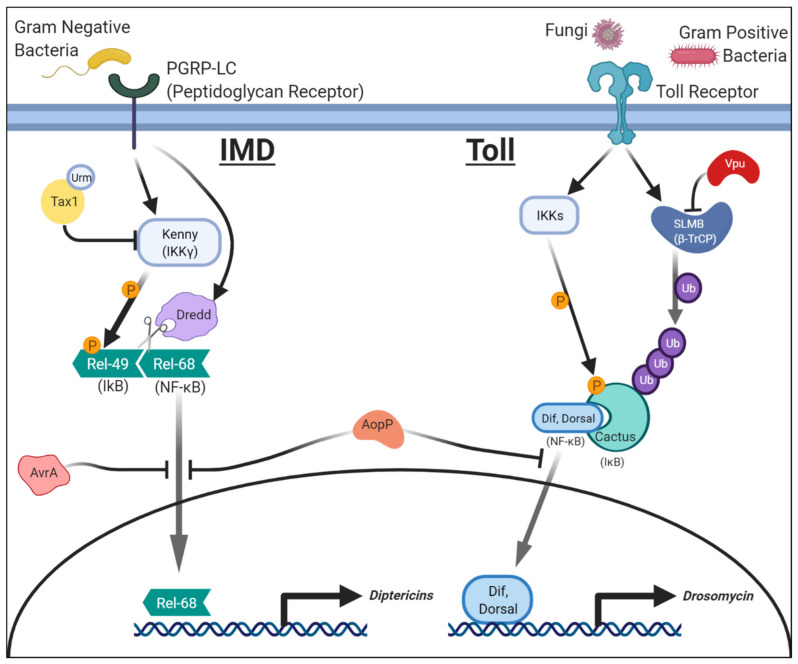
Virulence factors that affect immunological signaling pathways studied in *Drosophila*. Core components of the immune deficiency (IMD) (**left**) and Toll (**right**) branches of the NF-κB signaling pathway are depicted along with pathogenic proteins that target these pathways. Host proteins shown in cool colors and pathogenic proteins (AopP (*Aeromonas salmonicida*), and AvrA (*Salmonella enterica*), Tax1 (HTLV-1)) are shown in warm colors.

**Figure 2 ijms-22-02724-f002:**
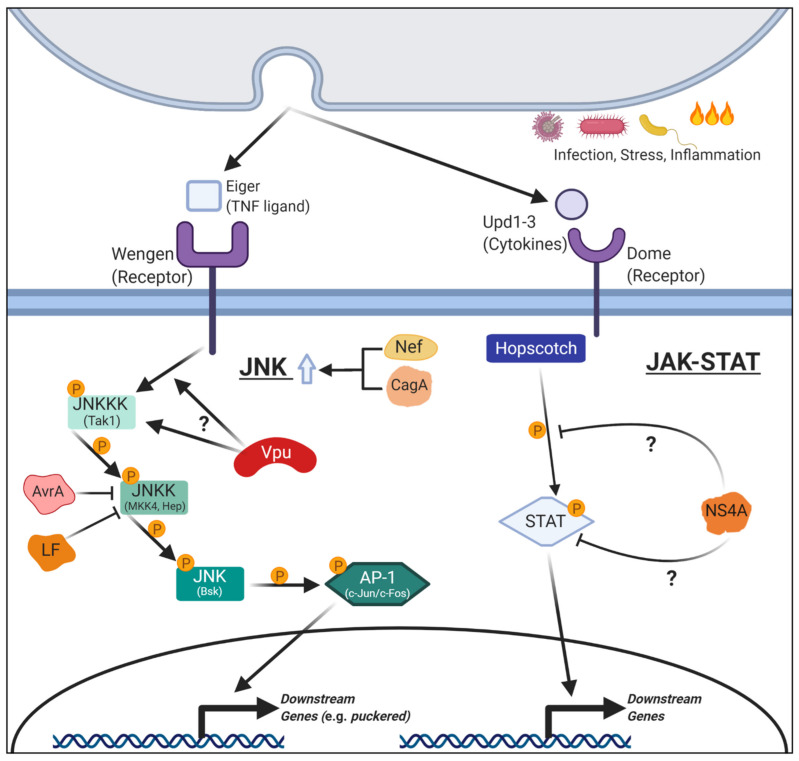
Virulence factors that affect stress-induced signaling pathways studied in *Drosophila*. Core components of the JNK (**left**) and JAK-STAT (**right**) pathways are depicted along with pathogenic proteins that target these pathways. Host proteins shown in cool colors and pathogenic proteins (AvrA (*S. enterica*), CagA (*Helicobacter pylori*), LF (*Bacillus anthracis*), Nef (HIV), NS4A (ZIKV), Tax1 (HTLV-1), and Vpu (HIV)) are shown in warm colors.

**Figure 3 ijms-22-02724-f003:**
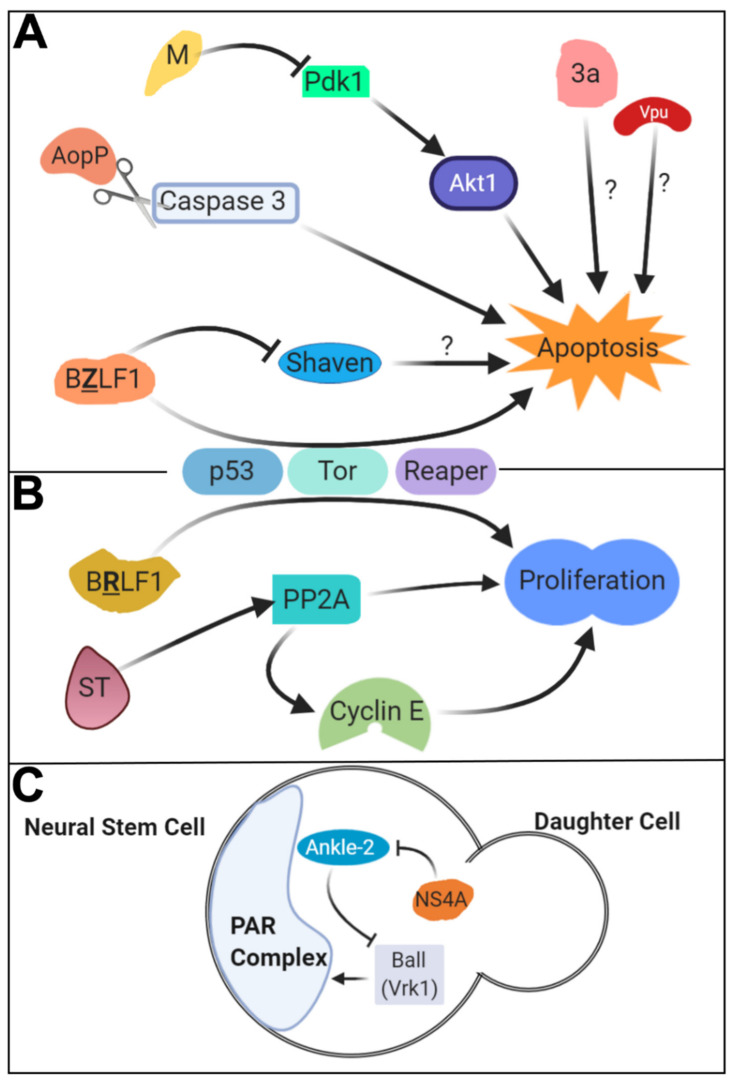
Virulence factors that affect apoptosis, cell proliferation, or asymmetric cell division studied in *Drosophila*. (**A**) Pathogenic proteins and their targets that affect apoptosis. Bacterial and viral proteins that induce apoptosis through activation of stress signaling pathways are not depicted here (see Figure 2 for these) (**B**) Pathogenic proteins and their targets that affect cell proliferation. (**C**) Pathogenic proteins and their targets that affect asymmetric cell division. Host proteins shown in cool colors and pathogenic proteins (3a (SARS-CoV-1), AopP (*A. salmonicida*), BRLF1 (EBV), BZLF1 (EBV), M (SARS-CoV-1), NS4A (ZIKV), ST (SV40), and Vpu (HIV)) are shown in warm colors.

**Figure 4 ijms-22-02724-f004:**
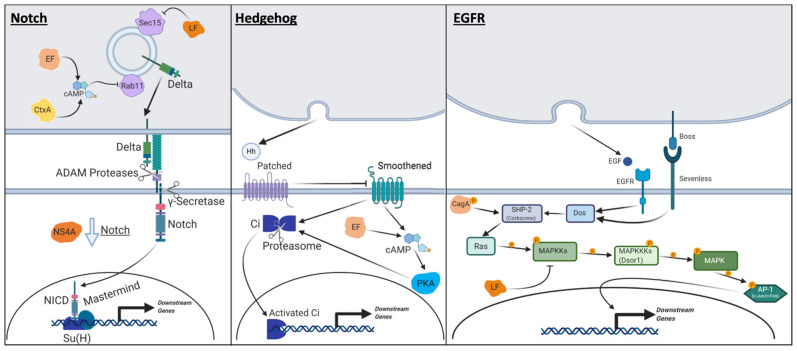
Virulence factors that affect developmental signaling pathways studied in *Drosophila*. Core components of the Notch (**left**), Hedgehog (**center**), and Ras-MAPK (**right**) pathways are depicted along with pathogenic proteins that target these pathways. Host proteins shown in cool colors and pathogenic proteins (CagA (*H. pylori*), CtxA (*Vibrio cholerae*), EF (*B. anthracis*), LF (*B. anthracis*), and NS4A (ZIKV)) are shown in warm colors.

**Figure 5 ijms-22-02724-f005:**
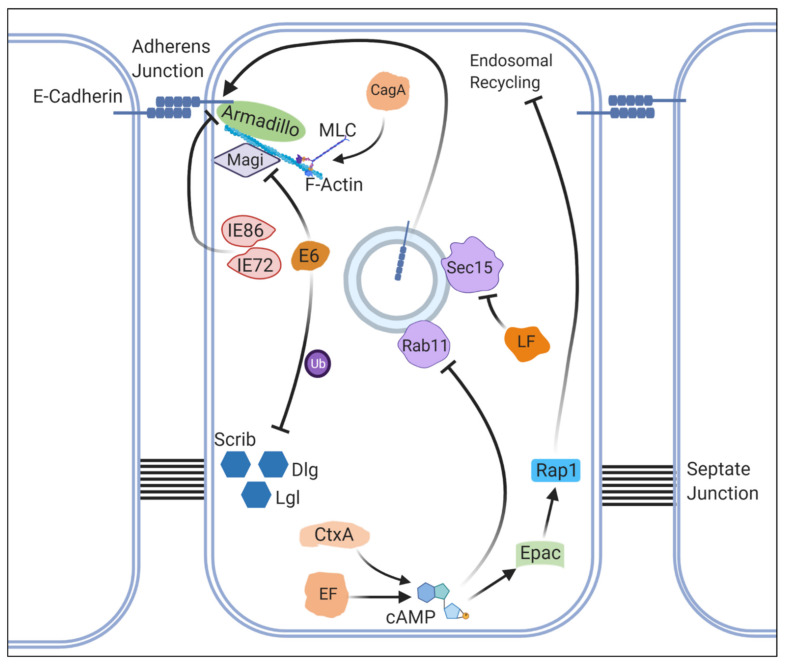
Virulence factors that affect cell adhesion or cytoskeletal proteins studied in *Drosophila*. Core proteins involved in cell adhesion are depicted along with pathogenic proteins that target these pathways. Host proteins shown in cool colors and pathogenic proteins (CagA (*H. pylori*), CtxA (*V. cholerae*), E6 (HPV), IE72/IE86 (HCMV), EF (*B. anthracis*), and LF (*B. anthracis*)) are shown in warm colors.

**Table 1 ijms-22-02724-t001:** Virulence factors studied in *Drosophila* and their host target proteins.

Type	Pathogens	Pathogenic Proteins	Section	Biological Processes Affected	*In vivo* Functions of Pathogenic Proteins
Bacteria	*Aeromonas salmonicida*	AopP	Section 2.2	NF-kB signaling	Blocks the nuclear translocation of NF-kB (Relish and DIF), inhibiting both IMD and Toll pathways.
			Section 2.2	Apoptosis	Facilitates the cleavage of Caspase-3, inducing apoptosis.
Bacteria	*Bacillus anthracis*	EF	Section 4.3	Hedgehog signaling	Hyperactivates PKA through its adenyl cyclase activity, activating Hedgehog signaling. Genetically interacts with *hedgehog*.
			Section 4.3	Notch signaling	Alters the subcellular localization of Delta ligands via affecting Rab11-dependent vesicle trafficking. Acts synergistically with LF protein.
			Section 4.3	Cell-cell adhesion	Alters the subcellular localization of E-Cadherin by activation of Epac through its adenyl cyclase activity.
		LF	Section 4.4	JNK signaling	Inhibits JNK signaling upstream of *hep* (JNKK) in the developing thorax.
			Section 4.4	EGFR signaling	Inhibits EGFR signaling in the developing wing disc through unknown mechanisms. Genetically interacts with *Dsor1* (MAPKKK).
			Section 4.4	Notch signaling	Alters the subcellular localization of Delta ligand via affecting Sec15-dependent vesicle trafficking. Acts synergistically with EF protein.
			Section 4.4	Cell-cell adhesion	Alters the subcellular localization of E-Cadherin.
Bacteria	*Helicobacter pylori*	CagA	Section 4.6	EGFR/Sevenless signaling	Activates EGFR signaling by mimicking the function of Dos (Gab-family protein) in a phosphorylation dependent manner through Corkscrew (SHP-2).
			Section 4.6	Cytoskeletal organization	Causes over-activation and altered subcellular localization of Spaghetti squash (Myosin light chain) via Rho GTPase in a phosphorylation-dependent manner.
			Section 4.6	JNK signaling and apoptosis	Activates JNK signaling upstream of Bsk (JNK), leading to increase in apoptosis.
			Section 4.6	Tumor metastasis	Synergizes with an oncogenic form of Ras (Ras^V12^) to facilitate the invasion of tumors formed in the eye. Genetically interacts with basolateral protein coding genes *dlg1* and *l(2)gl* that function as tumor suppressors.
			Section 4.6	Microbiome homeostasis	Causes dysbiosis of gastric microbiota when expressed in the digestive tract, leading to activation of immune responses in a phosphorylation-dependent manner.
Bacteria	*Pseudomonas aeruginosa*	ExoS	Section 3.1	Phagocytosis	Inhibits phagocytosis by blocking Rac2 (Rho family GTPase) function in hemocytes.
		gshA, gshB	Section 5.2	Bacterial stress resistance and biofilm production	Protects bacteria from ROS while negatively regulating the formation of biofilms.
Bacteria	*Salmonellae enterica*	AvrA	Section 2.1	NF-kB signaling	Blocks the nuclear translocation of NF-kB (Relish) in an enzymatic activity-dependent manner, inhibiting the IMD pathway.
			Section 2.1	JNK signaling	Decreases activity of MKK4 (JNKK), inhibiting JNK signaling.
Bacteria	*Vibrio cholerae*	CtxA	Section 4.4	Notch signaling	Alters the subcellular localization of Delta ligand via affecting Rab11-dependent vesicle trafficking in an adenyl cyclase activity-dependent manner.
			Section 4.4	Cell-cell adhesion	Alters the subcellular localization of E-Cadherin by affecting Rab11-dependent vesicle trafficking.
Virus	Epstein Barr virus (EBV)	BRLF1	Section 3.4	Cell proliferation	Promotes cell proliferation. Genetically interacts with *p53*, *Tor*, *reaper*, and other genes.
		BZLF1	Section 3.4	Apoptosis and cell proliferation	Works with *shaven* (Pax transcription factor) to facilitate apoptosis and inhibit cell proliferation. Genetically interacts with *p53*, *Tor*, *reaper*, and other genes.
Virus	Influenza virus	M2	Section 5.1	pH homeostasis	Increases intracellular pH through its function as a proton channel.
Virus	Human Cytomegalovirus (HCMV)	IE72, IE86	Section 4.5	Cell-cell adhesion	Alters the subcellular localization of Armadillo (b-Catenin).
Virus	Human Immunodeficiency Virus (HIV)	Nef	Section 2.5	JNK signaling and apoptosis	Activates JNK signaling, leading to an increase in apoptosis. Genetically interacts with *bsk* (JNK) and *hep* (JNKK).
		Tat	Section 4.8	Cytoskeletal organization	Decreases the rate of Tubulin polymerization during cytoplasmic streaming during oogenesis and mitosis during early embryogenesis.
			Section 4.8	Protein translation	Interferes with ribosome biosynthesis by binding to pre-rRNA and Fibrillarin.
		Vpu	Section 2.4	NF-kB signaling	Inhibits Slmb (b-TRcP) in a phosphorylation-dependent manner, activating the Toll pathway.
			Section 3.3	JNK signaling and apoptosis	Activates JNK signaling upstream of *hep* (JNKK) in a phosphorylation-independent manner, leading to an increase in apoptosis.
Virus	Human papillomavirus (HPV)	E6	Section 4.7	Cell adhesion and polarity	Causes disruption of cell adhesion and polarity by degrading proteins such as Dlg1, Scrib, and Magi with Ube3A (E3 ligase) during wing development.
			Section 4.7	Insulin signaling	Genetically interacts with a dominant-negative form of Insulin receptor during eye development.
			Section 4.7	Epithelial-to-mesenchymal transition (EMT)	Genetically interacts with oncogenic forms of Ras and Notch to contribute to EMT.
Virus	Human T Cell Lymphotropic Virus type 1 (HTLV-1)	Tax1	Section 2.3	NF-kB signaling	Inhibits Kenny (IKKg) in a Urmylation-dependent manner, activating the IMD pathway.
Virus	Severe Acute Respiratory Syndrome Coronavirus-1 (SARS-CoV-1)	3a	Section 3.2	PKB/AKT signaling and apoptosis	Inhibits the Pdk1-Akt1 axis of the PKB/AKT pathway, leading to an increase in apoptosis.
		M	Section 3.2	Apoptosis	Promotes apoptosis in an ion channel activity-dependent fashion.
Virus	Simian vacuolating virus 40 (SV40)	Small T antigen	Section 4.9	Mitosis	Causes mitotic spindle abnormalities by working with PP2A and upregulating Cyclin E expression.
Virus	Zika virus (ZIKV)	NS4A	Section 4.1	Asymmetric cell division	Inhibits Ball (Vrk1) to misregulate proper segregation of cell polarity regulators in neural stem cells.
			Section 4.1	Apoptosis	Induces apoptosis in the nervous system.
			Section 4.2	JAK-STAT signaling	Inhibits JAK-STAT signaling downstream of *hopscotch* (JAK kinase) in the developing wing.
			Section 4.2	Notch signaling	Inhibits Notch signaling in the developing wing through unknown mechanisms.

## Data Availability

Not applicable.

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
