# Peer review of "Drosophila as a Model for Infectious Diseases"

_ijms, 2021, doi:10.3390/ijms22052724_

Round 1

Reviewer 1 Report

This review article by Harnish et al highlights the utility of Drosophila melanogaster to study bacterial and viral infections.  The article thoroughly explains numerous strategies that have been used for this type of research and detail a number of novel insights into the pathogenesis of a wide array of pathogens.  It describes interactions between pathogen proteins and endogenous signaling pathways that were identified through elegant genetic screening strategies in Drosophila. The article makes a strong case for continued use of this “model genetic organism” in disease discovery and for using this system for new drug characterization.   Overall the writing is excellent, the figures illustrate the key ideas well, and the science is clear, so this is likely to be a well cited review. I have only minor comments to improve organization and clarity, especially for non-specialists, which I describe below.

Table 1 is extremely useful, but it either needs references or links to the details in the text for the original citations.

Line 53 – provide a definition of the innate immune system.

Line 59-60 – this sentence has a lot of jargon- not clear they need all of it, eg, FLP/FRT, CRISPRa/i, but the authors should define any terms at that point – maybe in a side bar if not in the main text.

Line 76- I disagree a bit with the statement that the “strategy is limited to pathogens that have the ability to infect Drosophila melanogaster” – there are numerous studies that have used organisms like E. coli, not a natural pathogen, and injected it into flies to determine the response. 

Section 2 is entitled “Pathogenic proteins that affect innate immune signaling pathways” but the section starts with describing the just pathways alone and the pathogens only come up at line 148- thus, a subsection title for this first part on the key pathways would be helpful to the reader.

(Related to line 101) Is there an analog in humans to the antimicrobial peptides produced in flies?

Figures - In general, I like the use of warm colors for pathogens, and cool colors for endogenous pathways, but there are a few places this is less obvious: Figure 1, the toll receptor is red and pathogen fungi is cool blue; in figure 4, Smoothened is red; in figure 5, actin is in red.

Some words seem to be omitted from the sentence starting in line 181 (“was not” appreciated?)

Figure 3 caption wording- would be clearer to say “pathogenic proteins and their targets that affect [process]” so it doesn’t sound like the pathogenic proteins affect the process and that the process has targets separately.  There is also an extra (B) in the caption.

Page 15/16 I wasn’t sure based on these descriptions if Urmylation of Tax1 was necessary for the inhibition of Kenny, or just the cytoplasmic localization? It is known?

Line 235-6 the wording is hard to follow and could be improved. In the second part, I wasn’t sure which component degradation was being referred to (Vpu specifically prevents the  degradation of the IB protein Cactus, the degradation of which prevents nuclear translocation of DIF)

Line 266 – is there evidence of a role for microtubules in plasmatocytes?  The examples later relate to actin, not microtubules.

Line 321- typo, unbiased

On Line 332 the authors cite a not-yet-peer-reviewed paper by Yang, S.; Tian, M.; Johnson, A. N.- does the review journal allow this to be cited?  If so, since it is more speculative, I wonder if it might fit better at the end of the review as a look to the future. It does make the interesting point about the ability to use this model to work out the mechanisms of new pathogens very rapidly, and is probably worth including if that is not restricted.

Given the description of Vpu on page 22, perhaps it would make sense to include Vpu with the apoptosis regulators in figure 3.

Line 424 delete “through”

Line 494- should refer to figure 4 (instead of figure 3)

Line 504 – should refer to figure 4 (instead of figure 3, although could include N as part of asymmetric cell division panel)

Line 509. “adherence” should be “adherens” (also in the figure)

Line 509, this description is a bit confusing since adherens junctions aren’t typically considered sufficient to create impermeable epithelial, this would usually be due to tight junctions/septate junctions.  I understand the authors are reporting on other people’s work, but I think this difference should be noted or it could be pointed out that it may be a multistep process.

Line 538- maybe “expressing” should be “expressed”

Line 540 – should refer to figure 4 (instead of figure 3)

Line 544 - should refer to figure 5, I think (instead of figure 4)

Given the description of CagA on page 36, the authors may consider including it and associated cytoskeletal components in figure 4. Or with its role in apoptosis, it may also be included in figure 3 with apoptotic regulators.

Line 651, although the original reference likely claimed there was metastasis, the authors here should clarify that overgrowing cells spreading to other tissues in Drosophila is somewhat distinct from metastasis that typically spreads through blood or lymph vessels.

The description on line 692 and further with E6 and UBE3A was a bit confusing. Are those the two proteins or are there 2 viral proteins?  If the UBE3A is a human protein in all the experiments, maybe the authors can denote it with an “h” (hUBE3A) in the text and figure to make this clearer.

With so many topics, it is hard to organize everything in a certain way and have it all fit.  There may be other orders for the subsections worth considering, and keeping the references to different figures together more. Or I wonder if it might be worthwhile to keep the viral sections together?  However, it is not a problem as it stands- there is not one obvious way to do this.

Section 5.2 provides interesting background to justify a drug screen in flies to combat Pseudomonas.  However, since such a screen itself is not described, the authors should be a bit more conservative about how they introduce the section, eg,  “a study that takes an infection-based approach to study the function of enzymes that contribute to biofilm formation in Pseudomonas, and how they [may] affect the bacteria’s sensitivity to certain types of drugs”

Author Response

>Reviewer 1:
>Comments and Suggestions for Authors
1. Table 1 is extremely useful, but it either needs references or links to the details in the text for the original citations.

We thank the reviewer for this suggestion. We added an additional column to Table 1 that refers to the specific section within the main text in which each protein is discussed. This will allow readers to easily identify the original citations.

  1. Line 53 – provide a definition of the innate immune system.

We added a brief description of the innate immune system as follows.

First, flies have an innate immune system that responds to foreign pathogens by activating cellular pathways to produce antimicrobial peptides, promote inflammation and recruit further immune system players including hemocytes that have phagocytic capacity.

  1. Line 59-60 – this sentence has a lot of jargon- not clear they need all of it, eg, FLP/FRT, CRISPRa/i, but the authors should define any terms at that point – maybe in a side bar if not in the main text.

We apologize for using several technical terms without detailed explanations. Because most of the terms listed here are not directly relevant to manuscripts discussed in this review, we decided to remove this jargon from our text. The readers can still look at the references associated with this statement [Citations 13-15] to read more about the breadth of technology that fly researchers have access to.

  1. Line 76- I disagree a bit with the statement that the “strategy is limited to pathogens that have the ability to infect Drosophila melanogaster” – there are numerous studies that have used organisms like E. coli, not a natural pathogen, and injected it into flies to determine the response.

We thank this reviewer for pointing this out, and indeed studies using non-infectious pathogens to flies such as E. coli has been used to study the Drosophila immune system. In the revised text, we clarified that one can use this methodology to study the pathogens that doesn’t necessarily infects flies in the wild as below.

“The benefit of this method is that it allows researchers to study host-pathogen interactions on a whole animal scale, and one can also study the effect of pathogens that does not infect Drosophila melanogaster in the wild.”

  1. Section 2 is entitled “Pathogenic proteins that affect innate immune signaling pathways” but the section starts with describing the just pathways alone and the pathogens only come up at line 148- thus, a subsection title for this first part on the key pathways would be helpful to the reader.

We renamed the title of this section to “Innate immune signaling pathways and pathogenic proteins that affect them” to reflect that we will be talking about the immune pathway first and then we will introduce the pathogenic proteins that affect these pathways.

  1. (Related to line 101) Is there an analog in humans to the antimicrobial peptides produced in flies?

Although there are no direct orthologs of fly AMPs in humans, we added a sentence and a citation that there are analogous genes in humans as following.

While there are no direct orthologs of Drosophila AMP genes in humans, a number of human peptides that have antimicrobial activity have been identified [29].

>Figures

  1. In general, I like the use of warm colors for pathogens, and cool colors for endogenous pathways, but there are a few places this is less obvious: Figure 1, the toll receptor is red and pathogen fungi is cool blue; in figure 4, Smoothened is red; in figure 5, actin is in red.

We thank the reviewer for pointing this out. We revised our Figures so that the color code scheme is consistent in the updated Figures.

  1. Some words seem to be omitted from the sentence starting in line 181 (“was not” appreciated?)

We have now corrected this typographical error.

>9. Figure 3 caption wording- would be clearer to say “pathogenic proteins and their targets that affect [process]” so it doesn’t sound like the pathogenic proteins affect the process and that the process has targets separately.  There is also an extra (B) in the caption.

We corrected the Figure 3 figure legend title as “Figure 3. Virulence factors that affect apoptosis, cell proliferation or asymmetric cell division studied in Drosophila.”, and further changed the subsection titles to “(A) Pathogenic proteins and their targets that affect apoptosis”, “(B) Pathogenic proteins and their targets that affect cell proliferation.”, “(C)Pathogenic proteins and their targets that affect asymmetric cell division.”. We also eliminated the extra (B) in this figure legend.

  1. Page 15/16 I wasn’t sure based on these descriptions if Urmylation of Tax1 was necessary for the inhibition of Kenny, or just the cytoplasmic localization? It is known?

Urmylation is required for proper subcellular localization of Tax1 (nuclear export). This process allows Tax1 to physically interact with Kenny, which is a cytoplasmic protein. To clarify this, we made a minor modification to the text as follows.

Based on these data, the authors proposed that Urmylation of Tax1 causes nuclear export of this protein, and this alteration in the subcellular localization facilitates the activation of the IMD branch of the NF-kB pathway by allowing Tax1 to interact with cytoplasmic proteins that regulate this pathway.”

  1. Line 235-6 the wording is hard to follow and could be improved. In the second part, I wasn’t sure which component degradation was being referred to (Vpu specifically prevents the  degradation of the IkB protein Cactus, the degradation of which prevents nuclear translocation of DIF)
    We rephased this sentence to clarify that the degradation of Cactus is necessary for nuclear translocation of Dorsal and DIF for Toll pathway activation, and that Vpu prevents the degradation. The revised section reads as follows.

Vpu specifically prevents the degradation of the IkB protein Cactus (Figure 1). Because Cactus degradation is required for the nuclear translocation of both DIF and Dorsal, Vpu inhibits the Toll signaling branch of the NF-kB pathway. Animals defective in Toll signaling cannot express Drosomycin in fat body cells, leading to immunodeficiency.

  1. Line 266 – is there evidence of a role for microtubules in plasmatocytes?  The examples later relate to actin, not microtubules.
    We apologize for this mistake and thank the reviewer for pointing this out. We changed “(e.g. microtubules)” to “(e.g. actin filaments) in this sentence.
  2. Line 321- typo, unbiased
    We have now corrected this typographical error.
  3. On Line 332 the authors cite a not-yet-peer-reviewed paper by Yang, S.; Tian, M.; Johnson, A. N.- does the review journal allow this to be cited?  If so, since it is more speculative, I wonder if it might fit better at the end of the review as a look to the future. It does make the interesting point about the ability to use this model to work out the mechanisms of new pathogens very rapidly, and is probably worth including if that is not restricted.

We confirmed with the editor that IJMS allows citation of preprint articles. Initially, we did consider mentioning this in the “6. Conclusions and Future Discussion” section but because this preprint is about the 3a protein of SARS-CoV-2, which is highly homologous to 3a protein of SARS-CoV-1 which is discussed in this section in depth, we felt it was most appropriate to mention it here. We would like to emphasize that we were very careful when citing article by adding the phrase “Although this manuscript needs to undergo peer review,…” so that the readers will not consider this as a paper that have undergone rigorous peer review.

  1. Given the description of Vpu on page 22, perhaps it would make sense to include Vpu with the apoptosis regulators in figure 3.

In the revised manuscript, we added Vpu into Figure 3 as a facilitator of apoptosis.

  1. Line 424 delete “through”

We have now corrected this typographical error.

  1. Line 494- should refer to figure 4 (instead of figure 3)
    We have now corrected this typographical error.
  2. Line 504 – should refer to figure 4 (instead of figure 3, although could include N as part of asymmetric cell division panel)

We changed “Figure 3” to “Figure 4”. Notch signaling indeed contributes to cell fate decisions following asymmetric cell divisions, but since it is not relevant to the studies discussed in our review, we decided to not include this information in Figure 3.

  1. Line 509. “adherence” should be “adherens” (also in the figure)
    We have now corrected this typographical error.
  2. Line 509, this description is a bit confusing since adherens junctions aren’t typically considered sufficient to create impermeable epithelial, this would usually be due to tight junctions/septate junctions.  I understand the authors are reporting on other people’s work, but I think this difference should be noted or it could be pointed out that it may be a multistep process.

We understand this reviewer’s argument that adherens junction is not directly involved in barrier function in epithelial cells. In Guichard et al., they did not look at septate junction (barrier forming junction in Drosophila) in their fly model nor did they look at tight junction (barrier forming junction in mammals) in their mouse model. To prevent any confusion, we rephrased this section as follows.

This was further shown to cause an increase in blood vessel permeability in mice, providing a molecular handle to begin to understand of one of the key pathogenic symptoms of anthrax, disruption of endothelial barrier integrity, for the first time [155].

  1. Line 538- maybe “expressing” should be “expressed”

We have now corrected this typographical error.

  1. Line 540 – should refer to figure 4 (instead of figure 3)

We have now corrected this typographical error.

  1. Line 544 - should refer to figure 5, I think (instead of figure 4)

We have now corrected this typographical error.

  1. Given the description of CagA on page 36, the authors may consider including it and associated cytoskeletal components in figure 4. Or with its role in apoptosis, it may also be included in figure 3 with apoptotic regulators.

In the revised Figures, we added CagA to Figure 5 based on this reviewer’s recommendation as a molecule that inhibits MLC, which is an Actin-based motor protein. To accommodate this change, we changed the title of Figure 5 to “Figure 5. Virulence factors that affect cell adhesion or cytoskeletal proteins studied in Drosophila.”. We decided not to include CagA as an apoptotic regulator in Figure 3 since this function has been proposed to depend on JNK signaling, which is already depicted in Figure 2.

  1. Line 651, although the original reference likely claimed there was metastasis, the authors here should clarify that overgrowing cells spreading to other tissues in Drosophila is somewhat distinct from metastasis that typically spreads through blood or lymph vessels.

We reworded this section based on this reviewer’s suggestion as follows.

Co-expressing CagA with RasV12 lead to a tumor that can spread into the fly central nervous system, demonstrating that CagA activation of the JNK pathway can synergize with oncogenic effect of hyperactive Ras signaling.

  1. The description on line 692 and further with E6 and UBE3A was a bit confusing. Are those the two proteins or are there 2 viral proteins?  If the UBE3A is a human protein in all the experiments, maybe the authors can denote it with an “h” (hUBE3A) in the text and figure to make this clearer.

E6 is a viral protein and UBE3A is a host protein. To avoid any confusions, we rephrased several sentences in this section as follows. We decided not use the ‘h’ to describe the human protein since it seems this style is not recommended based on the current nomenclature rules.

“E6 acts by binding to its substrates and directing them towards ubiquitin-mediated proteasomal degradation through the recruitment of the E3 ubiquitin ligase, UBE3A [a.k.a. E6 Adaptor Protein (E6-AP)] expressed by the host cell [220].”, “Transgenic flies capable of co-expressing HPV E6 and human UBE3A were developed to understand how E6 functions in vivo, and to further identify its targets [221].”, “While expressing E6 or UBE3A alone in the eye or wing did not cause any morphological defects, co-overexpression of the viral (E6) and host (UBE3A) proteins caused rough eyes and blistered/melanized wings.”, “Further assessment of the cellular consequence of co-overexpressing E6 and human UBE3A identified disruption of cell adhesion and polarity in the eye as well as excessive apoptosis in the wing primordium.”, “Experiments in the wing imaginal disc showed that while Dlg1, Scrib, and Magi can be degraded by co-expression of E6 and human UBE3A (Figure 5), p53 levels and subcellular localization does not change.”, “The authors also showed that developmental defects in the wing seen upon E6 and human UBE3A co-expression can be partially suppressed by co-overexpression of Magi, demonstrating Magi loss is at least partially responsible for wing phenotypes.”, “In addition, by performing a targeted genetic interaction screen focusing on signaling pathway genes, they found co-overexpression of E6 and human UBE3A alongside a dominant-negative form of Insulin receptor significantly worsened the eye phenotype caused by either E6 or human UBE3A alone, suggesting a synergistic interaction.”

>27. With so many topics, it is hard to organize everything in a certain way and have it all fit.  There may be other orders for the subsections worth considering, and keeping the references to different figures together more. Or I wonder if it might be worthwhile to keep the viral sections together?  However, it is not a problem as it stands- there is not one obvious way to do this.

We agree that this review covers many topics because we wanted this to be the most comprehensive review article about this specific topic so it will benefit readers from multiple fields. We tried many different ways to organize the different topics and papers, and we feel the current format is the most comprehensive in our opinion. We did attempt to group the viral sections and bacterial sections separately (as we did in Table 1) at one point, but considering some bacterial and viral proteins act on the same host protein or pathway, we felt it was more natural to group them based on the target host proteins and conserved cellular pathways these pathogenic factors act on. We acknowledge that the field of virology and microbiology is academically somewhat separated, but we hope a paper like this may stimulate the exchange of ideas across different fields.

  1. Section 5.2 provides interesting background to justify a drug screen in flies to combat Pseudomonas.  However, since such a screen itself is not described, the authors should be a bit more conservative about how they introduce the section, eg,  “a study that takes an infection-based approach to study the function of enzymes that contribute to biofilm formation in Pseudomonas, and how they [may] affect the bacteria’s sensitivity to certain types of drugs”

As this reviewer points out, the manuscript that is discussed here was not a drug screen paper. However, they were able to identify that Pseudomonas that carry specific mutations had increased sensitivity to ROS and less pathogenicity, knowledge that can be used to design specific drug screens towards different strains of Pseudomonas. To clarify this, we added the following sentence in the introduction of section 5.

“The second study took an infection-based approach to investigate the function of enzymes that contribute to biofilm formation in Pseudomonas, and how they affect the bacteria’s sensitivity to certain types of drugs. This work revealed that two genes, gshA and gshB, involved in glutathione synthesis change bacterial sensitivity to ROS, revealing a possible therapeutic avenue [258].”

Reviewer 2 Report

This manuscript represents a comprehensive and sophisticated overview of the literature concerning the application of the model organism Drosophila melanogaster in the research towards infectious human diseases. The manuscript is well organised and clearly written. The tables and figures are informative and illustrate the text in an appropriate fashion. This article will highlight the usefulness and successes in the application of Drosophila as a model for infectious diseases and its potential in understanding the underlying molecular and cellular mechanisms and in contributing to the development of new therapies and drugs. This article will also provide an important reference for the aim to refine, reduce and replace higher vertebrates as models for human diseases. For all of these reasons, I would like to congratulate the authors for their work and highly recommend publication of this manuscript.

some considerations for potential improvements:

(1) Figure 2. later in the manuscript the authors refer to Puckered-lacZ as a reporter for the JNK signaling pathway. The authors may want to include puckered into their cartoon in Fig. 2.

(2) The Gal4/UAS system is a key technique that is most frequently referred to in the manuscript and indeed may currently be the most versatile method for this type of research. Moreover, many of the studies referred to used the compound eye as a model. To make the review more accessible to non-fly readers, I would suggest that the authors include an additional figure to demonstrate how Gal4/UAS works and as an example to demonstrate how the GMR::Gal4 system can produce readily accessible phenotypes like a 'rough eye'. I am fully aware that this may sound a bit redundant to information present in the literature, but in the context of this article and concerning a larger, non-specialist readership, my sense is that it would help a lot.

(3) The authors have applied the term 'phenocopy' several times in the manuscript (p19, p33) to define similar phenotypes of genetic effects in different species such as human and fly. This is an incorrect use of the term. A phenocopy describes a similar/identical phenotypic effect in comparing a genetically based effect and an effect by a treatment such as chemical or physical treatment.

typos/missing words:

p13, line 181: sentence is incomplete

p14, header of figure 3: a verb is missing

p37, line 640: 'suggesting a synergistic relationship'

p40, line 716: 'a fly organ' instead 'an fly organ'

p42<, line 741: 'Polyomaviridae'

Author Response

>Reviewer 2:
1. Figure 2. later in the manuscript the authors refer to Puckered-lacZ as a reporter for the JNK signaling pathway. The authors may want to include puckered into their cartoon in Fig. 2.

We thank the reviewer for this suggestion. In the revised Figure 2, we added “puckered“ as an example of a downstream target gene of JNK signaling.

  1. The Gal4/UAS system is a key technique that is most frequently referred to in the manuscript and indeed may currently be the most versatile method for this type of research. Moreover, many of the studies referred to used the compound eye as a model. To make the review more accessible to non-fly readers, I would suggest that the authors include an additional figure to demonstrate how Gal4/UAS works and as an example to demonstrate how the GMR::Gal4 system can produce readily accessible phenotypes like a 'rough eye'. I am fully aware that this may sound a bit redundant to information present in the literature, but in the context of this article and concerning a larger, non-specialist readership, my sense is that it would help a lot.

We thank this reviewer’s suggestion. GAL4/UAS system is indeed the key technology that are used in most of the papers discussed in this review. While a graphical representation of this method may be useful, we instead decided to add a little more text to explain how this system works as follows. We explicitly state that the technology is described in detail in Brand and Perrimon (1993), which has an illustrated diagram. As this reviewer notes, similar images have been displayed in many review articles and text books.

For example, tissue and time specific expression of a specific gene, or groups of genes, can easily be performed using the UAS/GAL4 system, described in detail by Brand and Perrimon [12]. The UAS/GAL4 system allows controlled spatiotemporal expression of genes engineered with an upstream UAS (Upstream Activation Sequence) sequence. Flies harboring the UAS are then crossed to GAL4 lines in which the GAL4 is under the control of a specific gene promoter (e.g., GMR to drive expression in the developing eye). Since GAL4 activates the transcription of the gene downstream of UAS, any cell type in which the GAL4 is expressed will also express the gene under UAS control [12].

We also understand the benefit of showing a rough eye phenotype to non-Drosophila readers who have never seen this phenotype, but we decided not to include this since this is not directly related to our topic of interest. This phenotype along with other phenotypes like wing notching were used as functional readouts of specific signaling pathways, and we didn’t want this to be misinterpreted in other ways (e.g. defects caused by infection). We recommend the readers who are interested in learning more about these fly specific phenotypes and information to read the original manuscripts.

  1. The authors have applied the term 'phenocopy' several times in the manuscript (p19, p33) to define similar phenotypes of genetic effects in different species such as human and fly. This is an incorrect use of the term. A phenocopy describes a similar/identical phenotypic effect in comparing a genetically based effect and an effect by a treatment such as chemical or physical treatment.

We thank the reviewer for pointing this out. We changed the phrase “phenocopy” when we refer to analogous phenotypes that are seen in flies and human. The remaining “phenocopy” used in the text should be places where we refer to similar/identical effect in comparing a genetically different manipulation within Drosophila.

Together with Zika virus NS4A work discussed earlier [124,125], this work not only showcases the ability of fly researchers to study molecular functions of a pathogenic factor using Drosophila as a ‘living test tube’, but also highlights that flies can seemingly recapitulate human phenotypes for some infectious diseases, functioning as ‘preclinical disease models’.

>typos/missing words:
4. p13, line 181: sentence is incomplete

We have now corrected this typographical error.

  1. p14, header of figure 3: a verb is missing

We have now corrected this typographical error.

  1. p37, line 640: 'suggesting a synergistic relationship'

We have now corrected this typographical error.

  1. p40, line 716: 'a fly organ' instead 'an fly organ'

We have now corrected this typographical error.

  1. p42<, line 741: 'Polyomaviridae'

We have now corrected this typographical error.